# Micro-LED/van der Waals heterointegration for in-pixel processing display architecture

Fei Wang, Yuchun Wu, Hongling Chu, Jingbo Yang, Zhaorui Liu, Siqi Liu, Zhu Yang, Enlong Li, Jingjing Liu ⓘ, Luqiao Yin, Mengjiao Li ⓘ ✉ & Jianhua Zhang ✉

The rapid evolution of intelligent display technologies is driving the development of next-generation edge smart systems, yet conventional off-pixel processing architectures suffer from severe display latency and energy bottlenecks. Embedding in-memory computing features into pixel-level active drivers presents a promising strategy for co-locating image processing and display. In this work, we provide an in-pixel processed display-driven architecture that integrates micro-LEDs with $MoS_2$ in-memory transistors into a $16 \times 16$ active display array. Our in-pixel processing design exhibits high luminance ($>3 \times 10^5 \, cd \cdot m^{-2}$), high-speed operation (5000 Hz), and compact dimensions of $20 \times 35 \, \mu m$ per pixel. Crucially, leveraging the segmented voltage-luminance response and the non-volatile multilevel conductance update characteristics of the design devices, we demonstrate in-situ image reconstruction and real-time display using in-pixel processed micro-LED arrays. The image reconstruction capability of the proposed design is validated through a neural network-based image recognition task, where the inference process is implemented directly on the pixel array by deploying the trained weights via non-volatile conductance modulation. The reconstructed images achieved a significantly higher accuracy of 99.29% compared to the original inputs (79.81%). These results highlight the significance of the in-pixel processed display architecture as a promising approach to realize high-performance intelligent display technologies.

The rapid advancement of intelligent display technologies is revolutionizing interactive experiences and enabling next-generation Internet of Things applications[1–3]. Specifically, relying on typical silicon-based complementary metal oxide semiconductor (CMOS) driving circuits, transistor-capacitor cell design, and precise alignment bonding processes, micro-LEDs with high luminance and high resolution display features are propelling the blooming of intelligent display products[4–7]. However, with the increasing complexity and ever-growing loading for edge data processing escalating, conventional display-driving architectures relying on off-pixel processors are suffering from high data-shuttling latency and energy consumption[8–10]. This becomes a critical bottleneck for meeting high-performance display requirements in high-frame-rate imaging (120 Hz) and high-

dynamic-range rendering[11,12], necessitating a fundamental innovation in display driving design towards energy-efficient operation with integrated in-situ processing capabilities.

Image reconstruction, such as contrast and enhancement, as a pivotal stage in display pipelines, typically depends on discrete processing units (e.g., GPUs/ASICs) and complicated peripheral circuits[13–16]. Recent progress in in-memory computing architectures that empower low latency and parallel processing operations, offering a disruptive alternative for in-situ efficient display technology by embedding computational capabilities within active drivers[17–21]. For instance, emerging nonvolatile devices, such as floating-gate transistors, with analog switching behaviors can jointly drive micro-LEDs and perform in-situ processing[22,23]. Particularly, the linear voltage-

School of Microelectronics, Shanghai University, Shanghai, China. ✉e-mail: mjli@shu.edu.cn; jhzhang@oa.shu.edu.cn

luminance modulation enables low-level contrast calibration, while multistate charge storage facilitates high-level tasks like denoising and super-resolution reconstruction[24–26]. The original in-situ processed display concept was demonstrated in an electrolyte-gated synaptic transistor-driven quantum-dot light-emitting diode cell, achieving a luminance memory behavior, akin to the classical transistor-capacitor (e.g., 1T1C) driving design[27]. This scenario is subsequently extended to more compact light-emitting devices via an embedded conductance tuning layer, aiming to boost integration density and explore in-situ synaptic weight-dominated luminance modulation[28,29]. Despite these advances, current research on this aspect primarily focuses on either low-resolution low-luminance organic light emitting diode components or silicon-based driving circuits with fussy circuit design, low-yield bonding alignment, and high operating voltages. As the demand for ultra-high-resolution and reliable driving continues to rise, efficient intelligent display architectures call for collaborative optimization in driving configurations and mechanisms beyond silicon CMOS and oxide thin-film transistors.

The emergence of van der Waals (vdW)-based neuromorphic electronic devices, combining non-volatile memory and parallel processing characteristics, offers a high-density integrated driving platform ideally suited for micro-LED-based computational displays. Initial proof-of-concept studies, such as the monolithic integration of $MoS_2$ field-effect transistors driving micro-LEDs for high-luminance displays, have validated the potential of vdW devices in display driving areas[22,30,31]. To fully unlock the potential of vdW material for intelligent micro-LED displays, we pioneer an innovative in-pixel processed display-driven micro-LED display (IPPMLED) design with integrated in-situ processing capabilities. The proposed IPPMLED architecture integrates micro-LEDs with $16 \times 16$ $MoS_2$ synaptic transistor arrays as the pixel-level active drivers, achieving high luminance ($> 3 \times 10^5$ cd m$^{-2}$), a small pixel footprint ($20 \times 35$ μm), fast operating speed (5000 Hz), and operating reliability ($10^6$ cycles@120 Hz). Benefiting from the non-volatile, multilevel conductance update characteristics of the driver transistors, the flexibly modulated and temporarily stored luminance can be achieved in the integrated pixel devices, underlying their in-situ image processing and real-time display capabilities. Specifically, the piecewise linear voltage–luminance characteristics that closely align with pixel remapping schemes allow the implementation of the contrast stretching in the fabricated IPPMLED array, realizing the in-situ image enhancement display for noisy input letter images. By exploiting the pixel-level weight programmability of the IPPMLEDs, the trained neural network weights for image recognition can be deployed directly onto the IPPMLED arrays for hardware-accelerated inference. Consequently, the in-situ processed images achieve a recognition accuracy of 99.29%, significantly outperforming unprocessed inputs (79.81%). The integrated display approach in this work delivers a closed-loop solution that unifies image optimization and display within a compact, highly efficient pixel cell, advancing in-pixel image processing for next-generation intelligent display technologies.

## Results

### The structural design and device integration of IPPMLED
In contrast to traditional active-matrix micro-LED driver systems, where driver transistors merely function as current regulators for pixel switching and luminance control[32], the proposed IPPMLED design endows each driver transistor with parallel memory and processing capabilities, enabling efficient image reconstruction and real-time display. As illustrated in Fig. 1a, the IPPMLED architecture unifies the dual capabilities of in-situ image enhancement and on-chip weight updating for image recognition under a unified hardware platform. Specifically, by leveraging its gate-tunable luminescence properties, the device enables real-time contrast stretching (Fig. 1a(i)). This process dynamically remaps input pixel distributions for instantaneous

display. Simultaneously, the nonvolatile memory characteristics allow for the persistent retention of synaptic weights, thereby supporting the in-situ inference tasks required for robust image recognition via multiply-accumulate (MAC) operations within the IPPMLED-based pixel array (Fig. 1a(ii)).

Figure 1b shows the structural schematics of the designed IPPMLED display array and the corresponding manufacturing procedure. Each pixel unit comprises a vdW synaptic transistor and a micro-LED, forming a point-to-point modulation scheme via one-transistor-one-diode (1T1D) integration. Considering the high-resolution and high-yield requirements of future intelligent display technologies, a low-temperature-compatible maskless lithography process route is adopted in this study[33]. As illustrated in Fig. 1b, the main procedures include the precise patterning of the microscale electrodes, the definition of the monolayer vdW material $MoS_2$ active channels, the fabrication of the self-aligned bonding interfaces, and the low-temperature thermo-compression bonding (TCB) (<200 °C), leading to the vertical interconnection between the micro-LEDs and the driver transistor array. Specifically, the bottom contact configuration of the $MoS_2$-driver transistor enables rich modulation of the IPPMLED devices, which will be described in detail below. Note that 300 nm $SiO_2$ is employed as the gate dielectric during the early-stage exploration and validation of the proposed display architecture, owing to its advantages in better color contrast for transferred vdW materials and hence improving the processing reliability. Figure 1c provides the optical micrograph of the fabricated IPPMLED array, where each 1T1D cell integrates at the intersection of the gate and data lines to enable individual pixel addressing. The array features a pixel pitch of 84.75 μm, corresponding to an integration density of 300 pixels per inch (PPI). The clear surface image of an individual 1T1D pixel reveals the high fabrication quality of the designed processing route. Cross-sectional transmission electron microscopy (TEM) characterization with energy-dispersive X-ray spectroscopy (EDS) analysis examines the uniform interfaces and material composition of the vdW-driver transistors (Fig. 1d and Fig. S1a). Raman spectroscopy is further employed to confirm the crystalline property of the monolayer $MoS_2$ channels after TCB step, with typical $E^1_{2g}$ and $A_{1g}$ peaks observed at 383 cm$^{-1}$ and 408 cm$^{-1}$, respectively (Fig. S1b).

### Driving characteristics of IPPMLED
The electrical characteristics of the vdW-driver transistor serve as the critical determinant of pixel modulation in the IPPMLED array, directly impacting key display metrics such as luminance uniformity and stability. Figure 2a, b present the transfer and output characteristics of a representative $MoS_2$ driver transistor, respectively. It exhibits n-type behavior with good ohmic contact features[34]. The on/off current ratio exceeds $10^6$ orders ensures efficient current modulation and high display contrast. Notably, the observed clockwise hysteresis in the transfer curves manifests controllable charge-trapping dynamics during the carrier transport process[35], underlying the achievement of programmable luminance profiles in micro-LEDs through driving history manipulation.

Based on the efficient driving capability of the $MoS_2$ driver transistor, we further examine the electrical and luminescent characteristics of the integrated IPPMLED devices. As shown in Fig. 2c, the device-to-device variation is analyzed across 100 randomly selected IPPMLED pixels, for which the overlapped transfer characteristic curves with a variation of 11.98% recall the n-type conducting behavior of $MoS_2$ transistors in Fig. 2a. More statistical analysis of the threshold voltage ($V_{th}$), on-current ($I_{on}$), and off-current ($I_{off}$) is presented in Fig. 2e. Notably, variations among integrated units remain minimal with an average $I_{on}$ of $2.6 \times 10^{-6}$ A and $I_{off}$ of $5.5 \times 10^{-13}$ A. Figure 2d shows the output characteristic of the IPPMLED pixel, along with the corresponding light-emitting feature (red dashed curve). The turn-on data voltage ($V_{data}$) of IPPMLED can be identified as around 2.3 V, at

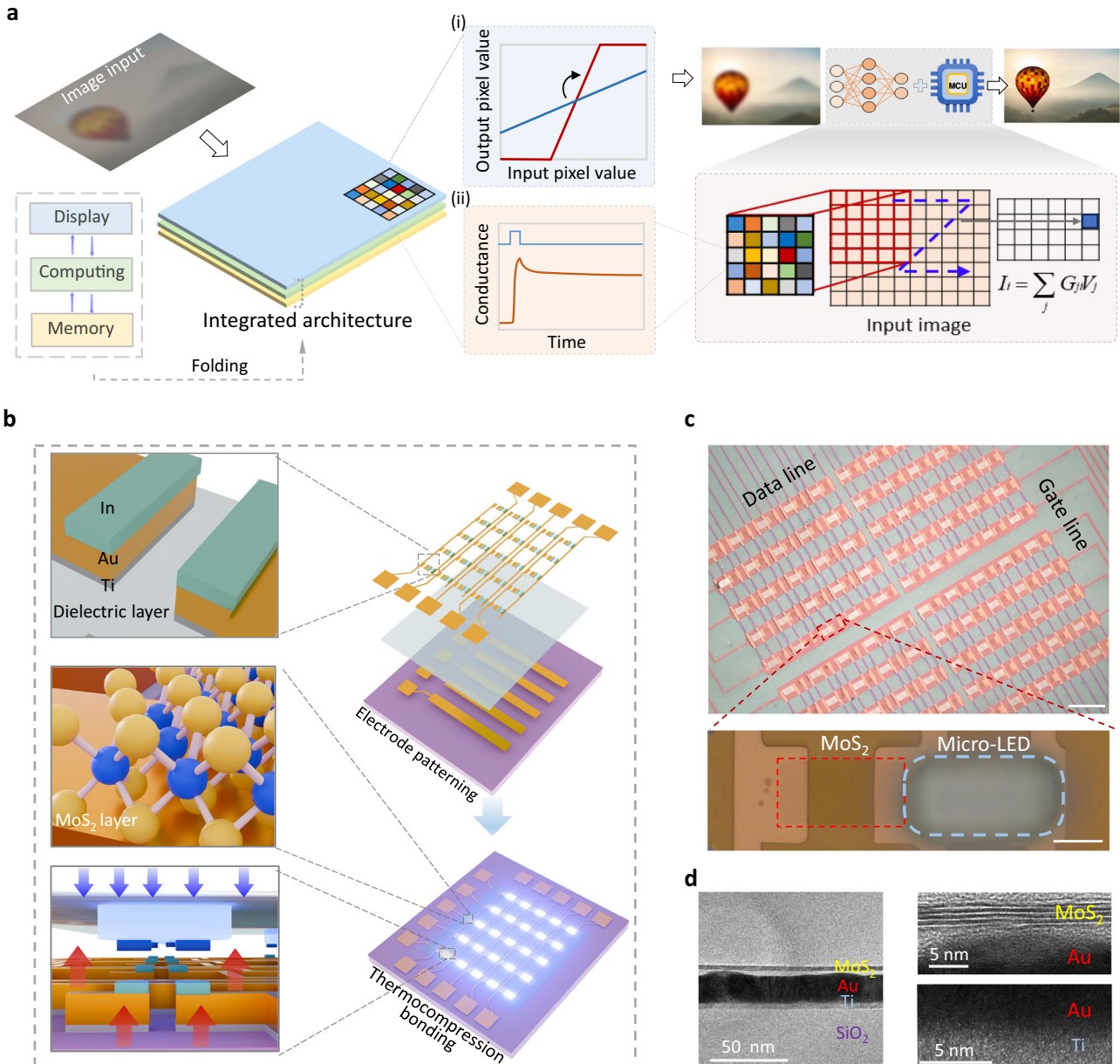

**Fig. 1 | Structural design and fabrication of the in-pixel processed display architecture. a** Schematic diagram of the in-pixel processed display-driven micro-LED (IPPMLED) architectures and functional implementation. The left panel illustrates the integrated memory–computing–display architecture, while the right panel highlights two in-situ image processing pathways: (i) hardware-level contrast enhancement via intrinsic voltage–luminance characteristics, and (ii) in-situ neural network inference by leveraging the non-volatile properties of IPPMLEDs for weight deployment and multiply-accumulate (MAC), with the microcontroller unit (MCU) serving as an auxiliary unit for voltage encoding and closed-loop optimization. The core MAC operation is expressed as $I_i = \sum_j G_{ji}V_j$, where $I_i$ is the total output current of

the i-th column, $G_{ji}$ represents the conductance of the transistor at the intersection of the j-th row and i-th column, and $V_j$ denotes the voltage applied to the j-th row. **b** Key steps in the fabrication process of the IPPMLED array, including electrode patterning, channel material transfer, and thermo-compression bonding, respectively. **c** Optical micrographs of the fabricated IPPMLED array (scale bar: 100 μm). Enlarged view provides the detailed one-transistor-one-diode (1T1D) pixel consisting of a micro-LED and a $MoS_2$ transistor driver (scale bar: 10 μm). **d** High-resolution cross-sectional transmission electron microscopy (TEM) image of the $MoS_2$ driver transistor, showing clear interfaces between the $MoS_2$ layer, metal contacts (Au/Ti), and $SiO_2$ dielectric layer.

which the micro-LED begins to emit visible blue light and then exhibits an exponential rise in luminance with respect to the $V_{data}$.

The blue-emitting micro-LED devices in this architecture are fabricated on the mature gallium nitride (GaN) platform (Supplementary Note 1 and Fig. S2), which offers significant advantages in material stability, electro-optical efficiency, and operational lifetime[36,37]. The micro-LED cell exhibits remarkable luminescence performance, achieving a peak external quantum efficiency (EQE) of 32.46% at a current of 0.29 mA, and a luminance of $3.13 \times 10^6$ cd m$^{-2}$ at a current density of $2.4 \times 10^5$ mA cm$^{-2}$ (Fig. S3). Fig. S4 provides the

electroluminescence (EL) characteristics of the micro-LED under various driving currents. The observed narrowband emission peaks and absence of spectral shift confirm the color purity and stable emission characteristics[38]. Such high-performance emission can be stably maintained over 2 h under a fixed driving voltage (Fig. 2f), and exhibits negligible luminance degradation over a 5-week duration (Fig. S5).

To evaluate the switching reliability of the integrated device, a pair of driving voltages is applied to alternately switch the IPPMLED pixels between the on state and the off state under a fixed $V_{data}$ of 3 V.

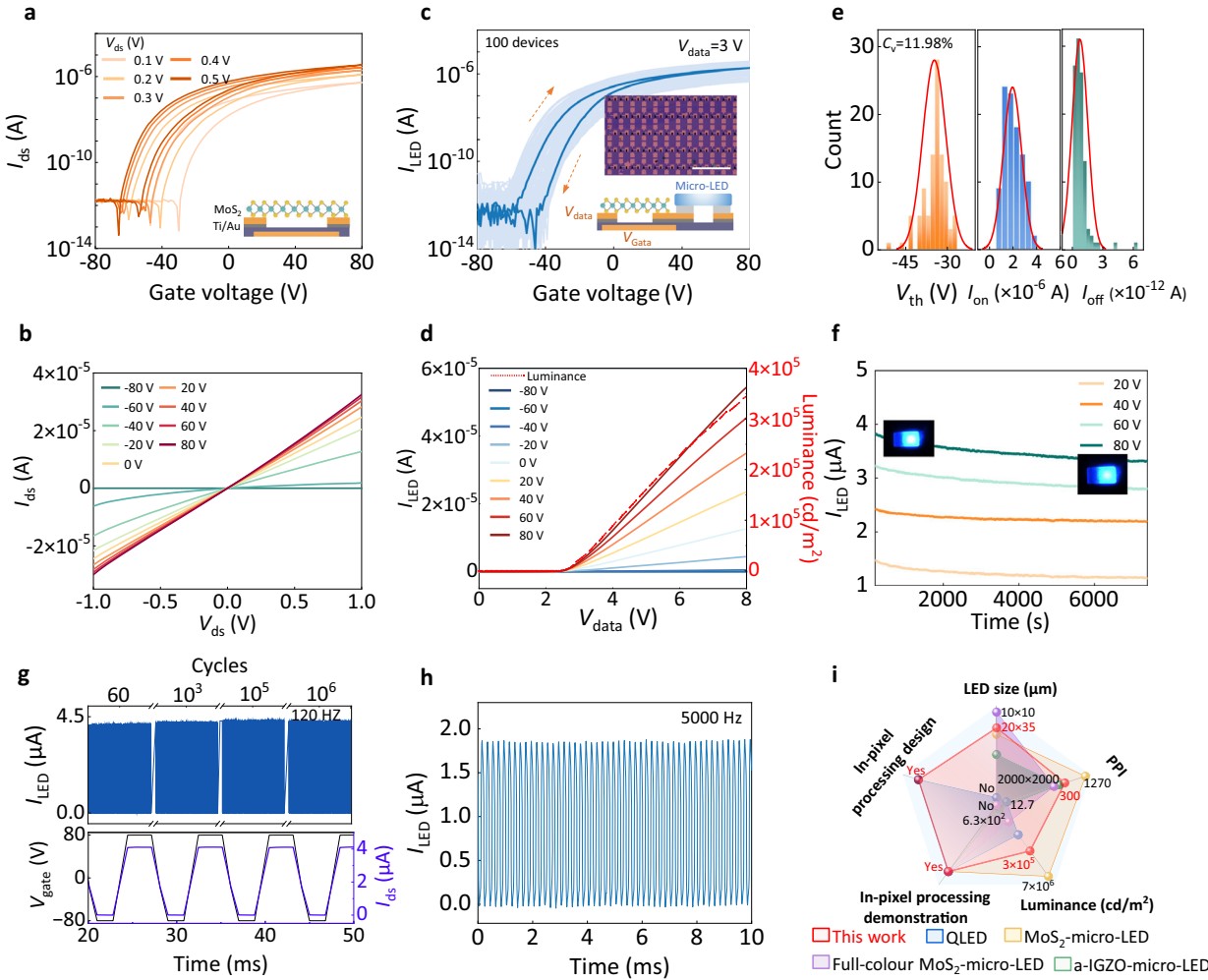

**Fig. 2 | Electrical characterization and performance benchmarking of the IPPMLED.** **a** Double-sweep transfer characteristics of a $MoS_2$ transistor under the double-sweep voltage $V_{ds}$ from 0.1 to 0.5 V. Inset shows the corresponding device schematic of $MoS_2$ transistor on $SiO_2$/Si substrate. **b** Output characteristics of the $MoS_2$ transistor. From bottom to top, the applied gate voltage ($V_{bg}$) varies from −80 to 80 V with a 20 V step. **c** Transfer characteristics of 100 individual 1T1D pixels at a data voltage ($V_{data}$) of 3 V, demonstrating high uniformity and reproducibility across devices. The darker solid line indicates the typical transfer curve. Insets show the optical microscope image of the IPPMLED cells (scale bar: 500 μm) and the corresponding device schematic of $MoS_2$ transistor-driven micro-LED. **d** Output current and corresponding light-emitting characteristics (dashed curve) of the IPPMLED pixel under different gate voltages. **e** Statistical distribution of threshold voltage ($V_{th}$), on-current ($I_{on}$), and off-current ($I_{off}$) from transfer characteristic curves at a fixed drain-source voltage of 3 V. The fitted curves obey the gaussian distribution of $f(G) = \frac{1}{\sqrt{2\pi}\sigma} e^{-\frac{(G-\mu)^2}{2\sigma^2}}$, where $\mu$ and $\sigma$ represent the mean and standard deviation, respectively. The device variation is characterized by the coefficient of variation ($C_v$), which is defined as the ratio of the standard deviation ($\sigma$) to the mean ($\mu$). **f** Stable multilevel emission of IPPMLED devices over 2 h. The insets show the luminescence states of the IPPMLED at various stages. **g** The stable operation of the IPPMLED device under 120-Hz driving pulses for over $10^6$ switching cycles. **h** Consistent switching behavior of the IPPMLED cell at a high frequency of 5000 Hz with a gate voltage swing of ±80 V. **i** Performance benchmark of the proposed IPPMLED devices with state-of-the-art display technologies, including LED size, pixels per inch (PPI), luminance, and in-pixel display processing capability[22,29,31,39].

The recorded current exhibits long-term endurance, maintaining stable and repeatable switching behavior over $10^6$ cycles at both 60 Hz and 120 Hz standard speed (Fig. 2g and Fig. S6), meeting the critical merits of device reliability and operational stability under continuous-use conditions. Furthermore, the IPPMLED also demonstrates consistent switching behavior over 10,000 cycles at a high frequency of 5000 Hz without obvious degradation (Fig. 2h and Fig. S7). Based on the obtained device performance, we further perform benchmark tests with existing active-driving display technologies, including luminance, pixel size, integration density, and switching characteristics (Supplementary Table 1). Figure 2i provides a clear comparison with the key metrics in the modern display fields[22,29,31,39]. Our IPPMLED design demonstrates a well-balanced display performance, particularly excelling in integrating in-pixel processing capabilities with practical applications. This integration underscores its potential for high-resolution and compact display implementations.

## Multilevel luminescence tuning of IPPMLEDs
Typical in-situ image presentation in display terminals, including image contrast and local pixel calibration, closely relies on controllable luminescence characteristics[40,41]. For example, the driving voltage-dependent luminescence behaviors dominate the contrast stretching principles. Thus, the driving voltage-dependent luminescence characteristics of the proposed IPPMLED devices are systematically investigated. Figure 3a exhibits the relationship between driving current and luminance of the IPPMLED pixel, revealing a strong linear evolution. This result indicates the luminance behavior is highly predictable, reducing the need for luminance compensation[12]. As the primary control parameter, gate voltage manipulates the emitted luminance behavior by regulating the injection current of the micro-LEDs. As shown in Fig. 3b, the gate voltage-luminescence dependency can be assigned as three phases: a non-emissive region at low gate voltages, a saturation region at high gate voltages, and a quasi-linear modulation

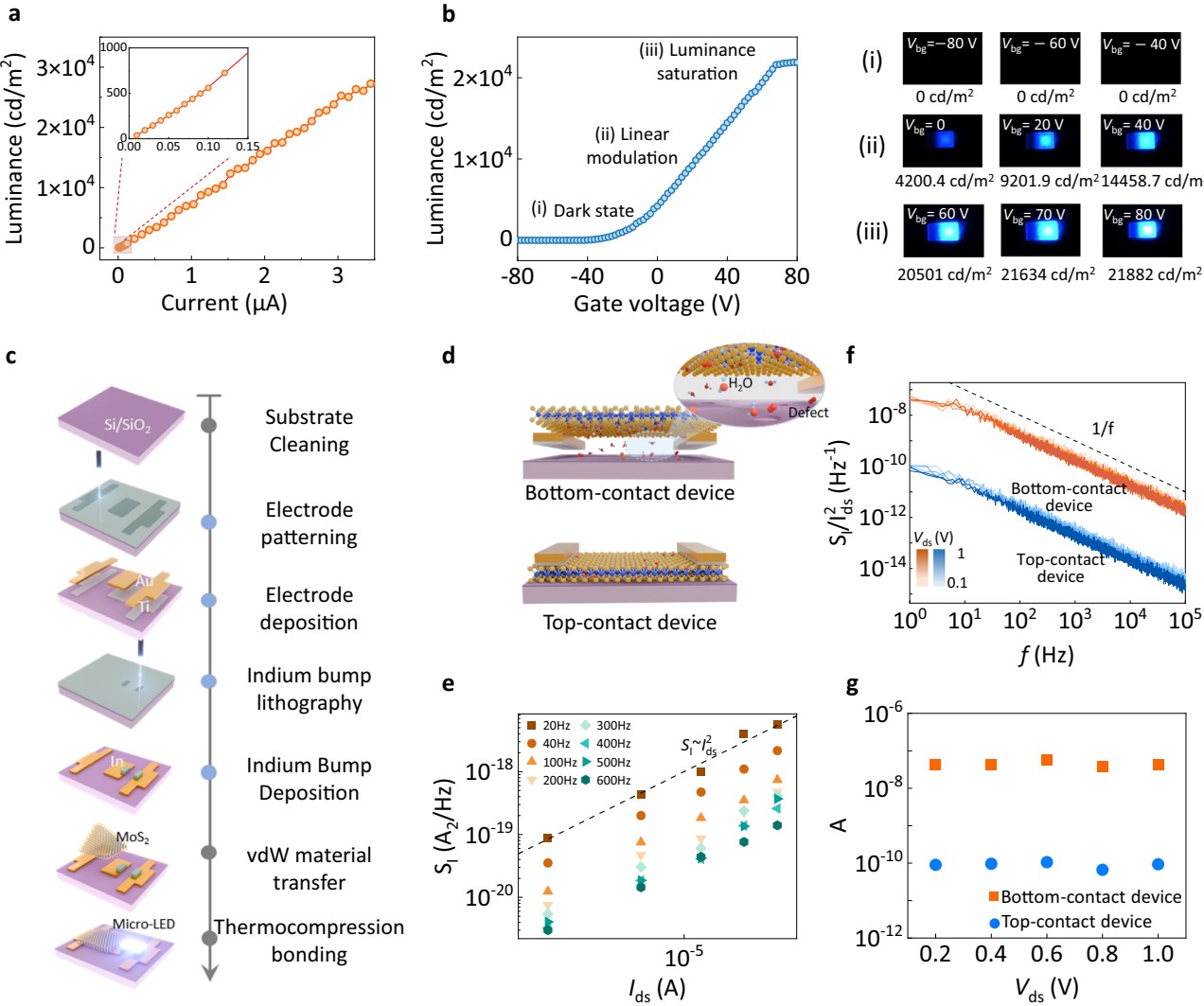

**Fig. 3 | Gate-tunable luminance features and mechanism analysis of IPPMLED.**
**a** The recorded current-luminance characteristic of IPPMLED devices, with an inset magnifying the 0–0.15 µA region. **b** Gate voltage–dependent luminance characteristics and the corresponding light emission behaviors of IPPMLED devices under different stages. **c** Schematic illustration of the fabrication process for back-gate bottom-contact IPPMLED devices. **d** Structural schematics of different device designs for bottom-contact and top-contact configurations. **e** The measured current noise power spectral density ($S_I$) as a function of drain-source current ($I_{ds}$) at different frequencies. The dashed line shows $S_I \cdot I_{ds}^2$. **f** Drain current normalized $S_I$ depending on $I_{ds}$ of bottom-contact device and top-contact device at several drain-source voltages. A dashed line with a slope of −1 is included as a reference for ideal 1/f noise. **g** The extracted noise amplitude ($A$) of devices at different drain-source voltages ranging from 0.2 to 1 V.

region in the middle, where luminance can be precisely and predictably controlled. This segmented luminance modulation characteristic provides a fundamental principle for contrast enhancement in display systems, enabling per-pixel luminance remapping through gate voltage encoding.

Besides, higher-quality image presentation necessitates advanced high-level image reconstruction, where embedded intelligent calibration at the display pixel terminals is expected[42–45]. In this context, IPPMLED devices serve as pixel-level weight updating units for neural network–based image processing, physically embodied by the programmable and nonvolatile conductance states. This necessitates an in-depth investigation into the weight plasticity dynamics for shaping the pixel-level in-situ processing capability. We first review the uniform charge-trapping transport behavior in Fig. 2a, for which the uniform charge-trapping transport behavior is critical in shaping synaptic plasticity. Its physical origin can be understood by the bottom-contact fabrication procedures in Fig. 3c, during which the patterning of source/drain electrodes is followed by the channel transfer process. This scenario, compared to the typical top-contact configuration

(Fig. S8), would provide a broader space, i.e., vdW gaps, for residual adsorbates (such as $H_2O$ and oxygen) to be trapped at the interface of the active channel and the substrate (Fig. 3d)[46,47].

Furthermore, both microscopic observation and holistic dynamic electrical characterizations are performed to validate this hypothesis. A high-resolution cross-sectional TEM image reveals the suspended structure between $MoS_2$ and $SiO_2$, in good agreement with the measured electrical characteristics (Fig. S9). Low-frequency noise analysis for two devices with top and bottom configurations further allows us to visualize the dynamic charge trapping-detrapping mechanisms by diagnosing the carrier transport fluctuations[48,49]. The measured power spectral density (PSD) of the current fluctuations ($S_I$) for both devices exhibits an ideal 1/f behavior, indicating the uniform distribution of charge fluctuations (Fig. S10). Figure 3e shows the PSD as a function of drain-source current ($I_{ds}$) of bottom-contact device. The PSD scales quadratically with $I_{ds}$ across frequencies, confirming 1/f noise generation governed by carrier-number fluctuations in the channel. Subsequently, the normalized PSDs ($S_I/I_{ds}^2$) of both bottom-contact and top-contact devices are analyzed at different drain-source voltage ($V_{ds}$)

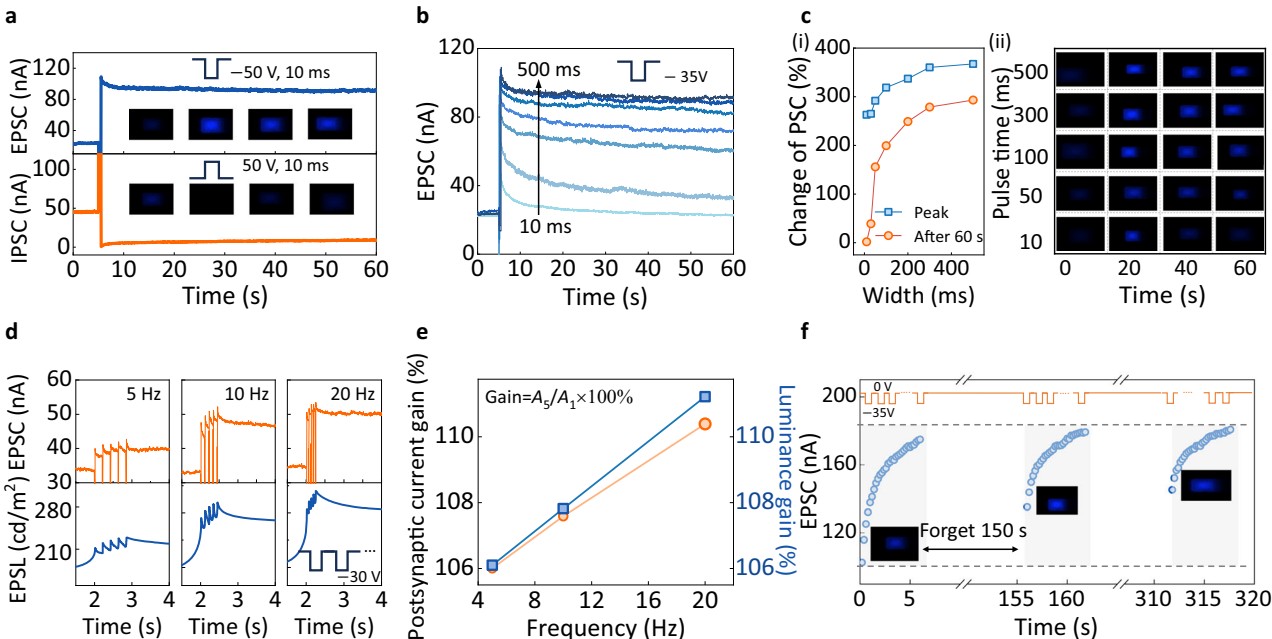

**Fig. 4 | Non-volatile luminescence characteristics of micro-LEDs driven by synaptic transistors. a** Synaptic plasticity-based luminescence behaviors of the IPPMLED devices, including excitatory postsynaptic current (EPSC) and inhibitory postsynaptic current (IPSC), along with visual micro-LED luminescence modulation (inset images). **b** Long-term luminescence characteristics of the IPPMLED under different pulse widths, demonstrating multilevel brightness retention for analog signal encoding. **c** EPSC response of the IPPMLED devices under different pulse widths (i), accompanied by the corresponding micro-LED luminance variation (ii). The change of PSC is defined as $\triangle I/I$, where $\triangle I$ is the current amplitude shift after the pulse and $I$ is the current before the pulse. **d** The recorded excitatory postsynaptic luminance (EPSL) of the IPPMLED device under increasing driving-pulse frequencies. **e** Gradual enhancement of current gain in the driver transistor and luminance gain in the micro-LED as a function of driving-pulse frequency. **f** Learning-forgetting-relearning behavior of the IPPMLED devices, illustrated by the accumulation of EPSC after 30 consecutive pulses followed by a 150-s delay. The inset shows the corresponding gradual enhancement in the micro-LED's emission intensity. The horizontal dashed lines delineate the range of device current variation.

values, as presented in Fig. 3f. Following the semiempirical equation of $S_I/I_{ds}^2 = A/f$, a critical parameter of noise amplitude ($A$) can be extracted, which marks the level of the charge fluctuation behaviors. As expected, bottom-contact devices exhibit significantly higher $A$ values, confirming the critical role of heterogeneous interfacial defects in carrier modulation under the bottom-contact process (Fig. 3g).

The elucidated physical mechanism of the charge transport behavior in driver transistors propels us to explore the weight-update-dominated luminescence characteristics for achieving the in-situ processing capability in the IPPMLED devices. As shown in Fig. 4a, we probe synaptic-shaped display luminance evolution in individual pixels by applying different polarities of the gate-voltage pulse in the driver transistor. Specifically, under a positive pulse, electron accumulation occurs on the surface of the active channel, causing the energy levels to bend downward and enabling trap states below the Fermi level to be filled. While under a negative pulse, the trapped carriers are released, making the traps empty[50]. Thus, the recorded excitatory postsynaptic current (EPSC) and inhibitory postsynaptic current (IPSC) behavior reshape the light emission of the micro-LED, giving rise to long memory luminescence and transient quenching behaviors in the IPPMLED cell, named excitatory postsynaptic luminance (EPSL) and inhibitory postsynaptic luminance (IPSL), respectively. The insets of Fig. 4a present the complete temporal luminescence profile of EPSL and IPSL phenomena. Notably, the memory luminescence characteristics of EPSL provide a solution for eliminating the inherent blanking intervals in refresh cycles, effectively replacing the pixel-refresh circuitry used in traditional active matrix displays—such as the 1T1C design for maintaining the pixel-driving voltage across the storage capacitor component. This capability critically addresses the need for enhanced visual smoothness in modern intelligent display technologies.

To systematically investigate the multilevel modulation behavior of pixel-level weight updating, a diverse range of driving situations is considered. As illustrated in Fig. 4b, the recorded EPSC curves following the increase in pulse width correspond to a multilevel memory function, which is visualized by the luminance density duration of micro-LEDs. Figure 4c depicts the extracted evolution of EPSC and the corresponding luminance modulation trajectories of micro-LED under varying pulse widths. The characteristic multilevel luminance dynamics, initiated from a dark state, reaching maximum luminance, and subsequently undergoing gradual dimming, clearly demonstrate the pronounced influence of pulse widths on the conductance-dependent luminescence characteristics. This scenario can be reproduced under varying pulse amplitudes (Fig. S11), indicating precise electrical modulation of device weights.

The conductance-dependent luminance modulation behavior under diverse driving conditions is further examined. For example, paired-pulse facilitation luminance, akin to the typical feature of short-term synaptic plasticity, is emulated by applying successive driving gate spikes (Fig. S12). It is found that the PPF index decreases gradually with increasing interval time ($\Delta t$), following a double exponential function. Given that the refresh frequency is a critical metric for display technologies, driving frequency-dependent EPSL behavior is further investigated. Figure 4d shows that the EPSC and EPSL of the device as a function of driving frequency increase steadily from 5 to 20 Hz. The extracted EPSC gain can reach above 110% at 20 Hz, with the EPSL demonstrating a comparable gain level (Fig. 4e). Figure 4f demonstrates the learning–experience characteristics of the integrated IPPMLED cell under a stimulation protocol of 30 consecutive electrical pulses followed by a 150-s recovery period. Specifically, the EPSC response is shown to increase, surpassing the previous baseline levels, as evidenced by the progressively intensifying light output in the inset

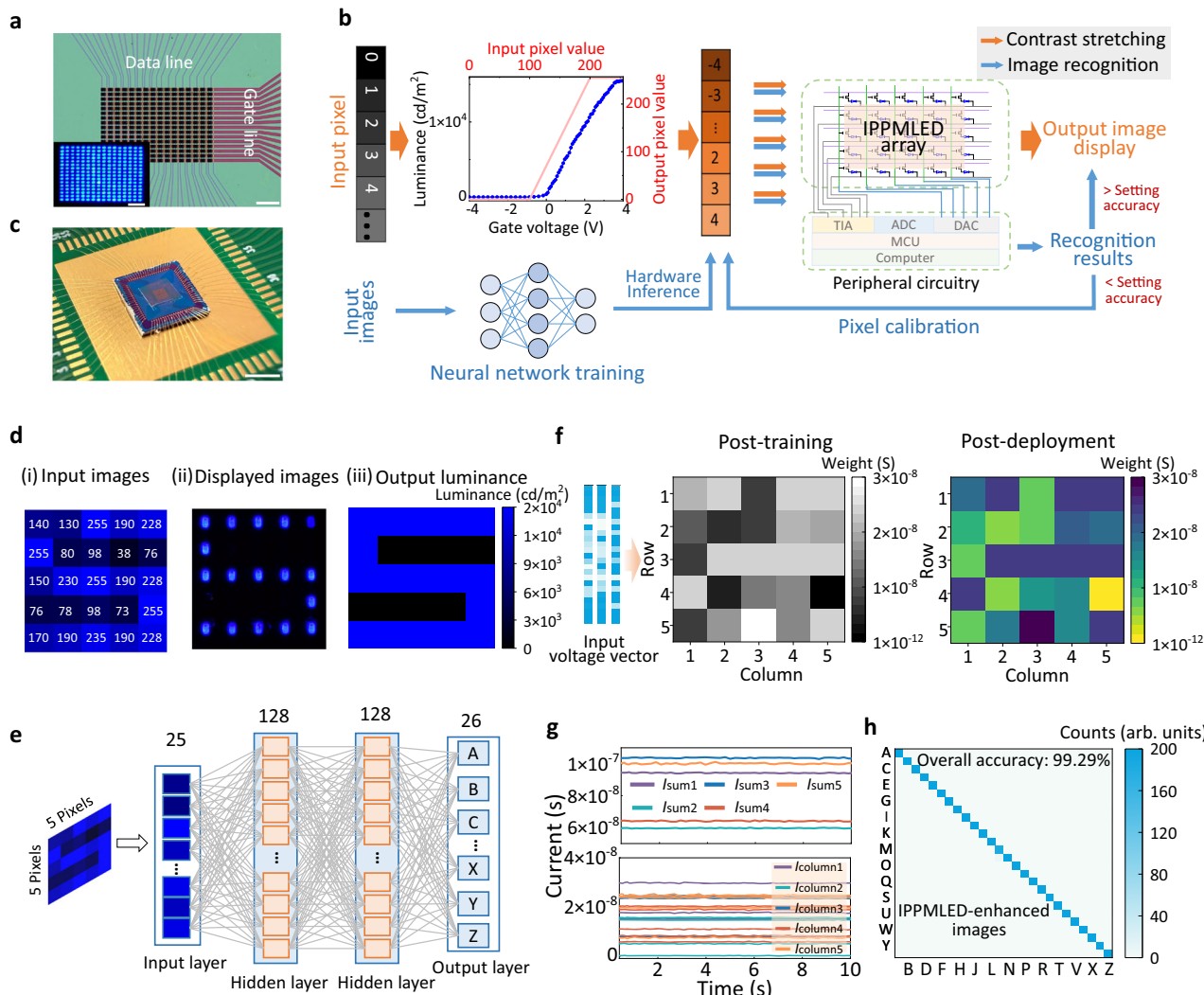

**Fig. 5 | Demonstration of the in-pixel image enhancement and recognition based on IPPMLED arrays. a** Optical micrographs of the fabricated $16 \times 16$ IPPMLED array with HfO$_2$ film serving as the dielectric layer (scale bar: 200 μm). The inset shows the uniform light emission behavior across the entire array (scale bar: 200 μm). **b** Schematic illustration of the contrast stretching and image recognition flow charts of the proposed in-pixel display processing architecture. The associated peripheral circuitry comprises a transimpedance amplifier (TIA), analog-to-digital converter (ADC), digital-to-analog converter (DAC), and MCU. **c** Wire bonding of the HfO$_2$-based IPPMLED array on a printed circuit board (scale bar: 5 mm). **d** Input image of noisy characters (i), the IPPMLED-enhanced image (ii), and the output

luminance map (iii). **e** Schematic of the feedforward neural network for image recognition, comprising a three-layer fully connected architecture with two 128-neuron hidden layers and a 26-class output layer. **f** Weight deployment and hardware inference process. Input images ($5 \times 5$ pixels) are flattened into $25 \times 1$ vectors, split into five sub-vectors, and sequentially processed via multiply-accumulate operations with the mapped $5 \times 5$ weights on the array. Heatmaps show the hardware conductance distribution (post-deployment) versus the quantized software weights (post-training). **g** The column current measurements of an IPPMLED array during the MAC process. **h** Confusion matrix showing the recognition results for the hardware-based inference performed on the IPPMLED array.

image. In summary, the IPPMLED device exhibits diverse conductance plasticity characteristics under different pulse conditions, enabling versatile display operating modes.

On the other hand, to meet the stringent power constraints of edge computing applications, we systematically explored the potential of high-κ dielectrics in replacing SiO$_2$ gate insulator for power consumption optimization[51,52]. HfO$_2$, as a widely adopted high-κ dielectric material, exhibits good insulating properties and favorable interface characteristics, which are compatible with the designed fabrication process flow of the IPPMLEDs. Figure 5a shows the successfully fabricated $16 \times 16$ HfO$_2$-based IPPMLED arrays with an increased pixel density of 336 PPI by reducing the pixel pitch. The detailed process optimization for large-scale array integration is provided in Fig. S13 and Fig. S14, showcasing refined methods for large-area vdW material transfer and micro-LED bonding techniques. As expected, the IPPMLED device with dielectric engineering exhibits a notable

reduction in the driving voltage, while maintaining good device-to-device uniformity in both $V_{th}$ and on/off-state across 100 devices ($C_v = 9.9\%$) (Fig. S15). Correspondingly, the inherently low power consumption of the synaptic transistor translates into a great energy efficiency advantage for the integrated LED device over current display technologies (Fig. S16 and Supplementary Table 2).

Similarly, ideal nonlinear or triplet-stage voltage-dependent luminescence characteristics along with non-volatile memory behavior can be reproduced in the IPPMLED device with HfO$_2$ dielectric engineering. As illustrated in Fig. S17a, when the gate voltage is swept from −4 to +4 V, the device exhibits three well-defined emission regions, characterized by a non-emissive dead zone below 0 V, an approximately linear response regime, and finally a saturation region at intermediate to higher voltages (above 3 V). Furthermore, under programming pulses within a low voltage window of ± 4 V, the device exhibits controllable conductance modulation, mirroring the versatile

synaptic plasticity essential for in-memory computing (Fig. S17b–e). Specifically, the conductance state demonstrates non-volatile conductance retention lasting over 150 s with wider-pulse stimulus, highlighting its potential for neural network weight deployment applications. In contrast, the device achieves instantaneous refresh responses under short pulse widths, such as at frequencies of 60 Hz, 120 Hz, and 240 Hz, emphasizing its capability for display refreshing (Fig. S18). Moreover, the HfO$_2$-based IPPMLED demonstrates robust endurance, maintaining stable operation over $10^4$ cycles (Fig. S19). This reliable cycling endurance is critical for subsequent practical in-situ display applications that require long-term device stability.

### IPPMLED-enabled in-situ display processing design

The blooming of intelligent edge technologies, especially for edge-oriented visual platforms, demands stringent requirements on efficient and in-situ image data processing capabilities. Conventional display architectures, however, typically offload a variety of tasks—ranging from low-level image processing (e.g., contrast stretching, gamma correction) to high-level functions (e.g., recognition, deblurring)—to external processors[53]. This physical separation between the processor and the display terminal leads to significant reconstruction latency and extra energy waste, generally causing noticeable visual dizziness[54,55]. In this work, the IPPMLED features gate-tunable luminance response and in-memory computing capability, offering significant advantages for in-situ display processing. To fully harness the advantages of the proposed devices, both contrast stretching and image recognition are taken as classic examples to exhibit how the IPPMLED array provides a processing platform towards an optimal balance between in-pixel processing and off-chip management.

To comprehensively assess the capability of the proposed IPPMLED in reshaping the intelligent in-situ display landscape, a synchronized display processing pipeline capable of handling representative multi-level tasks has been designed. As shown in Fig. 5b, the input image pixels are first converted into driving voltages according to the device-specific voltage–luminance characteristic, which intrinsically aligns with the contrast-stretching algorithm. In this scheme, low pixel values are mapped to the sub-threshold voltage region (no emission), high values to the saturation region, and intermediate values to the linearly stretched operating region, respectively, thereby enabling immediate contrast-enhanced display. To further optimize the display output, an in-pixel recognition function is proposed. Specifically, the IPPMLED array, serving as the neural network accelerator, enables efficient hardware inference by executing batched weight deployment and in-situ MAC operations. The system subsequently determines whether to trigger MCU-assisted image deblurring according to the obtained recognition results. For instance, if the obtained recognition accuracy falls below a predefined threshold, an advanced deblurring procedure would be activated with the assistance of the MCU (STM32F103RCT6) and back-end processors. Conversely, a higher recognition accuracy would bypass high-level processing to achieve the high-quality display.

### In-situ display processing tasks with IPPMLED arrays

To demonstrate the practical in-situ image processing and display capability of the designed IPPMLED, we have established a hardware platform that integrates the IPPMLED array with peripheral circuits to perform in-pixel processing and real-time image display, following the synchronized display processing pipeline (Fig. S20). In specific, each pixel in the IPPMELD array is equipped with independently addressable data and gate lines to enable precise control over input and output terminals (Fig. 5c). The successful operation of the wire-bonded array on a custom-designed printed circuit board (PCB) is demonstrated in Fig. S21, which exhibits uniform pixel illumination across the array.

First of all, the fundamental contrast stretching capability of the fabricated IPPMLED array is investigated. The demonstration employs letter patterns S, H, and U with noticeable noise pixels as the original input images. Specifically, the pixel values of the input image are converted to corresponding voltage signals spanning from −4 to 4 V and injected into the IPPMLED array. The detailed pixel-to-voltage mapping during the contrast stretching procedure is provided in Supplementary Note 2 and Fig. S22. Each pixel element autonomously processes its input voltage based on intrinsic voltage-dependent luminance characteristics, providing real-time optical feedback. Distinctly, with the device-specific contrast stretching, the input images are enhanced by the IPPMLED array that suppresses noise pixels and boosts feature pixels, effectively yielding clear images and operating steadily (Fig. 5d).

The quantitative assessment is further considered to validate the contrast stretching operation, providing better guidance for subsequent refinement of the display architecture. We demonstrate an in-pixel image recognition using an IPPMLED-based feedforward neural network, applied to both original noisy letter images and their IPPMLED-enhanced counterparts (Fig. 5e). As shown in Fig. 5f, the weights obtained from ex-situ software training are extracted, quantized, and then programmed onto the IPPMLED array via non-volatile conductive states, achieving a complete correspondence with the target weight matrix. Subsequently, the letter images ($5 \times 5$ pixels) are transformed into voltage vectors to serve as inputs, where parallel MAC operations are performed with the deployed weights, enabling highly parallel, low-power analog computing and in-situ execution of the inference process within the IPPMLED. Figure 5g illustrates a representative current summation process across the array during this operation. The resulting MAC values are temporarily stored in the readout circuit and buffer registers, after which weight refreshing and array reuse enable the processing of subsequent batches, ultimately yielding the recognition result. As expected, the IPPMLED-enhanced images demonstrated a high inference accuracy of 99.29%, outperforming the unprocessed images (79.81%) (Fig. 5h and Fig. S23), enabling high-quality display output while eliminating the need for further processing. Notably, the hardware inference accuracies exhibit only a minor degradation of 0.1–0.9% compared to their software counterparts (Fig. S24), which strongly validates the functional reliability of the IPPMLED-based in-situ image display system.

## Discussion

In summary, we propose a computation-in-pixel architecture named IPPMLED that seamlessly integrates micro-LEDs with vdW in-memory computing transistors. Through the advanced flip-chip bonding process, this integrated device achieves great performance metrics, including high luminance ($>3 \times 10^5$ cd m$^{-2}$), compact pixel dimensions of $20 \times 35$ μm, a switching frequency exceeding 5000 Hz, and an endurance of over $10^6$ switching cycles. Based on the voltage–luminance modulation properties and non-volatile conductance updating characteristics of the driver transistors, a hardware platform centered around the IPPMLED array has been constructed to achieve integrated in-situ image processing and real-time display. Specifically, the fabricated IPPMLED array demonstrates image enhancement of noisy inputs (S, H, and U) through piecewise linear voltage–luminance characteristics that closely match pixel remapping schemes. An IPPMLED-based neural network hardware further implements an in-pixel image recognition task for high-quality image deblurring (99.29%) through non-volatile weight deployment with the aid of peripheral control capability. The compact in-pixel display processing design in this work establishes a critical pathway toward next-generation immersive interfaces by merging display functionality and efficient processing at edge intelligent architectures.

## Methods

### Device fabrication

The GaN-based blue LEDs are fabricated beginning with the metal-organic chemical vapor deposition (CVD) growth of n-GaN, InGaN/GaN multiple quantum well, and p-GaN epitaxial layers on sapphire substrates. The mesa structure is then defined by inductively coupled plasma etching to expose the n-GaN layer, followed by sputtering and patterning of an indium tin oxide current spreading layer. An Ag reflective layer is subsequently deposited via electron-beam evaporation to enhance light extraction efficiency. Device isolation is achieved through plasma-enhanced CVD of a $SiO_2$ layer with via openings formed by dry etching. Finally, patterned metal electrodes are fabricated by electron-beam evaporation to complete the pixel array. A detailed flowchart of the fabrication process is provided in Fig. S2.

To fabricate the IPPMLED array, local gate terminals are first patterned using a laser direct writing (LDW) lithography system. Ti/Au gate metals with thicknesses of 5/30 nm are then deposited via electron-beam evaporation, where Ti serves as an adhesion layer to improve Au-substrate bonding and form ohmic contact with the semiconductor channel. A $SiO_2$ dielectric layer with a thickness of 300 nm is subsequently fabricated by CVD. Then, bottom-contact source and drain electrodes are fabricated, followed by the location of a $MoS_2$ film (Six Carbon Technology) and channel patterning via a wet-transfer technique and plasma etching. The indium (In) electrode pattern is then defined by LDW, and a low-pressure e-beam evaporation process is used to deposit the In layer (1.6 μm), forming a self-aligned interface for subsequent micro-LED integration. Finally, TCB is employed to precisely bond micro-LEDs onto the In bumps, completing the flip-chip bonding process. Furthermore, this fabrication scheme is successfully extended to fabricate IPPMLED arrays featuring an $HfO_2$ dielectric layer. A uniform 20 nm $HfO_2$ film was deposited by atomic layer deposition, a technique selected owing to its great conformality and precise thickness control at the atomic scale.

### Structural analysis and electrical characterization

The surface morphology of the samples is captured using a Focused Ion Beam-Field Emission Electron Microscope (FIB–FESEM, Helios G4 UX), which is also used for sample preparation for TEM. High-resolution imaging and element analysis are performed on the samples using a High-Resolution Transmission Electron Microscope (JEM-2100F) coupled with an Energy Dispersive X-ray Spectroscopy (EDS) system. Electrical performance testing of the devices is conducted using a Semiconductor Parameter Analyzer (Keysight B1500A equipped with a WGFMU). For Raman spectroscopy, a continuous-wave laser (MSL-U-532, 532 nm) is used as the excitation source, and a Raman spectroscopy system is set up with an HRS300 spectrometer and an 1800 lines/mm diffraction grating to collect and analyze the Raman signals from the samples. The optical performance of the micro-LEDs is further characterized by measuring the luminance with a COLOR VISION SV-2000 optical testing system and the EQE with an EVERFINE AIS-2_0.5 m integrating sphere, respectively.

### Training and inference of the IPPMLED-based neural network

A multilayer perceptron is developed for character classification using a custom dataset of 5 × 5 grayscale letter images. The model adopts a three-layer, fully connected architecture, which includes two hidden layers with 128 neurons each and an output layer corresponding to the 26 character categories. The total number of trainable parameters of the entire neural network amounts to 23194. A Rectified Linear Unit activation function follows each hidden layer to enhance the model's nonlinear representational capacity. The network is trained for 250 epochs using the AdamW optimizer, with the cross-entropy loss serving as the optimization objective. To evaluate generalization performance, the dataset is divided into separate training and validation subsets, and validation accuracy is recorded after each

epoch. The dataset comprises 26,000 grayscale images across 26 classes (A–Z), with 1000 samples per class. Each character is represented as a 5 × 5 pixel array. To improve robustness against input perturbations, controlled-amplitude random noise is applied for data augmentation. All image samples are normalized prior to being fed into the network.

For hardware-accelerated neural network inference, an integrated system comprising an $HfO_2$-based IPPMLED array, peripheral electronics (INA226A ADC, DAC7311 DAC), and a microcontroller unit (MCU, STM32F103RCT6) is developed. The $HfO_2$-based IPPMLED array is configured in a matrix layout and wire-bonded onto a PCB (ASM Pacific Technology, IHAWK AERO). Programming voltages within the range of ± 4 V are supplied by an NI-PXI 4163 modular instrumentation system, integrated with a 24-channel PXI-2532B multiplexer for row and column addressing. During inference, the trained synaptic weights are deployed onto the IPPMLED array via non-volatile conductance modulation. Each input image (5 × 5 pixels) is flattened into a 25 × 1 vector, divided into five sub-vectors, and sequentially encoded into voltage signals proportional to pixel intensity for in-situ MAC operations with the mapped 5 × 5 weights on the array. The resulting output currents are then amplified and converted by peripheral readout circuitry, with the processed signals being temporarily stored in buffer registers. Following this, the array undergoes weight refreshing and is reused to process subsequent image batches, ultimately generating the final recognition result.

## Data availability

Relevant data supporting the key findings of this study are available within the article and the Supplementary Information file. All raw data generated during the current study are available from the corresponding authors upon request.

## Code availability

The codes used for simulation and data plotting are available from the corresponding authors upon request.

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

## Acknowledgements

The authors would like to acknowledge the financial support provided by the National Key Research and Development (R&D) Program of the Ministry of Science and Technology (2024YFA1211500) and the National Natural Science Foundation of China (62304320 and 62574123).

## Author contributions

M.L. and F.W. conceived and designed the experiments. F.W. fabricated the device and F.W., H.C., and J.Y. conducted electrical measurements. J.L. and Y.W. performed the simulation of the artificial neural network. F.W., S.L., H.C., and E.L. completed the data analysis together. The paper is written by M.L. and H.C. with contributions from all the co-authors. Z.L., Z.Y., and L.Y. provided some experimental methods. M.L. and J.Z. supervised the research. All the authors discussed the results and commented on the manuscript.

## Competing interests

The authors declare no competing interests.
