## [Transparent Peer Review file · Nature Communications]

Micro-LED/van der Waals heterointegration for In-pixel processing display architecture

Corresponding Author: Professor Mengjiao Li

Version 0:

Reviewer comments:

Reviewer #1

(Remarks to the Author)

This study introduces an in-pixel processed micro-LED display (IPPMLED) architecture that integrates MoS₂-based synaptic transistors with micro-LEDs for combined pixel driving and memory-related functionality. The system demonstrates luminance levels above 3×10^5 cd/m², integration density of 14,000 units/cm², and operation up to 5000 Hz, with reported cycling stability. Using voltage-luminance modulation and multilevel conductance characteristics, the device is applied to image enhancement and MCU-assisted deblurring. A proof-of-concept 5×5 matrix is presented, showing noise suppression and image reconstruction, with recognition accuracy on the Fashion-MNIST dataset improving from 83.79% to 96.31%. While these results suggest the potential of combining display and processing functions within individual pixels, the demonstrations are limited in scale, and further work is needed to verify scalability, power efficiency, and stability in practical contexts. The concept is of interest, but revisions are necessary to clarify the scope of the hardware contribution and address methodological issues. The detailed comments are as follows:

1. The devices operate at relatively high gate voltages due to the use of a 300 nm SiO₂ dielectric, which raises concerns regarding energy efficiency for the portable applications (e.g., AR/VR systems). Although the supplementary information briefly mentions HfO₂ as a possible replacement to lower the operating voltage, the manuscript does not provide a systematic analysis of overall power consumption. For targeting edge computing applications, such quantitative evaluation is essential to support the claimed applicability.
2. In Figure 4f, the authors demonstrate potentiation behavior through output current changes under repeated pulses, illustrating a learning-forgetting-relearning cycle. While this result confirms short-term synaptic plasticity, the analysis is limited to tens of seconds. For establishing robust memory functionality, additional data on long-term retention and potentiation–depression characteristics (linearity, symmetry) are necessary.
3. A central claim of the manuscript is that the device enables hardware-level contrast stretching by exploiting its piecewise voltage-luminance response. However, the detailed explanation of this process, including the mapping relationship and supporting histograms, is presented only in the Supplementary Information. Given its importance to the overall contribution, this description should be included in the main text to substantiate the in-situ processing claims. Otherwise, the role of hardware in the contrast stretching demonstration remains unclear.
4. The description of the deblurring demonstration lacks sufficient detail. Although the manuscript states that the process is “MCU-assisted,” it does not clearly explain which parts are carried out by the device and which by the MCU. Moreover, the algorithmic details are missing. It is not specified whether a neural network was used, what network architecture or parameters were adopted, or how the computation was divided between the MCU and the device. Without this information, it is difficult to determine whether the demonstration truly reflects in-pixel processing or is largely dependent on external computation. A clearer and more rigorous explanation of the deblurring methodology is therefore essential.

5. Across the demonstrations, it is unclear whether true in-situ processing was actually carried out at the hardware level using MoS₂ synaptic transistors. The descriptions of contrast stretching and MCU-assisted deblurring do not provide sufficient evidence that these functions were intrinsically realized within the device rather than externally emulated. For the central claim to be convincing, the authors should present a more precise explanation of how in-situ processing was implemented, along with clear experimental evidence demonstrating that the computation indeed occurred in the hardware.

6. Figure 1a is described simply as a “schematic diagram of the IPPMLED architectures and functional implementation,” but the caption lacks sufficient detail to help readers interpret the figure. The right-hand side shows an image processing part that is only described as “functional implementation” in the caption. However, it is unclear how image processing operations are mapped onto the pixel-level hardware. Also, the label “ADMLED” appears, but this abbreviation is never defined in the manuscript. Since this schematic diagram is crucial to illustrate the computational aspect of architecture, the authors should provide a clearer explanation of the processing pathway and its correspondence with the device functions.

7. In Figure 2e, the average I_{on} is described in the text as $\sim 2.6 \times 10^{-6} A$, but the figure label appears to indicate $I_{on} (\times 10^6 A)$, which is misleading. Similar typographical errors (e.g., fig.3b, “matrixes” instead of “matrices”) are present. Careful revision is needed to avoid confusion.

Reviewer #2

(Remarks to the Author)

This manuscript presents a novel in-pixel processed display-driven micro-LED display (IPPMLED) architecture that integrates micro-LEDs with molybdenum disulfide (MoS₂) in-memory computing transistors. The goal of this architecture is to overcome the latency and energy consumption bottlenecks of conventional off-pixel processing by co-locating image processing and display at the pixel level. The authors fabricated a 5x5 IPPMLED matrix and demonstrated its capabilities in both low-level and high-level image processing.

The reviewer believes that the following revisions will help improve the quality of the manuscript.

Comment #1: The authors' claims about “high-resolution display” are ambitious given that the hardware demonstration is limited to a 5 x 5 matrix. Wafer-scale, high-uniformity MoS₂ TFT backplanes remain challenging (many processes still rely on flake transfer or small-area CVD growth), whereas micro-LEDs are already proven at very high pixel densities. This creates a scalability tension between the transistor backplane and the emitter array. While the reported pixel density of 300 PPI is notable, please provide a credible scaling roadmap and demonstrate a larger matrix to validate the claim.

Comment #2: The current array performs contrast enhancement task, but practical image processing pipelines also require low-level (e.g., color correction, gamma correction), mid-level (e.g., segmentation, feature extraction), and high-level (e.g., recognition) processing. Please clarify how these additional functions would be handled by the in-pixel architecture versus off chip.

Comment #3: The main limitation of trap-based synaptic devices is slow processing, which may cause image ghosting. In Fig. 4a-c, the postsynaptic current appears to stabilize over tens of seconds, which conflicts with high refresh rate operation. Please report the relevant time constants at the operating point and demonstrate video-rate updates without image persistence. If the display mode does not rely on slow synaptic effects, please state this explicitly and provide measurements that confirm millisecond scale settling suitable for typical display rates of 60 to 240 Hz.

Comment #4: Please expand the micro-LED section to include fabrication flow and key device metrics such as luminance, current density, and EQE to clarify emitter performance.

Comment #5: In Figure 2e, the variation of threshold voltage appears large for display-class uniformity, which could limit high-resolution operation. Please describe compensation methods and pathways to enhance the uniformity.

Comment #6: Only blue micro-LED devices are demonstrated. Please discuss how the in-pixel processing systems scale to multi-color RGB displays.

Comment #7: In Fig. 3b, it seems that the required gate voltages are quite high. Is this common, and realistic for actual display operation? If not, are there ways/roadmaps to reduce this?

Comment #8: The demonstrated pixel density of ~ 300 PPI is insufficient for microdisplays which typically require much higher PPIs (>2000) to reduce screen door effects. Therefore, the references to microdisplays in the abstract and introduction seem a bit misleading.

Comment #9: In Table S1, I think it could be better to also display the integration density in PPIs since it's more commonly used for displays than units/cm².

Comment #10: The authors should provide a more thorough analysis of the device's power consumption, comparing that of IPPMLED to conventional off-pixel display system. This would provide a more complete picture of the energy efficiency

benefits.

Reviewer #3

(Remarks to the Author)

Version 1:

Reviewer comments:

Reviewer #1

(Remarks to the Author)

The revised manuscript addresses many of the earlier concerns, with improvements in overall organization, clearer figure annotations, and added benchmarking context. Nonetheless, several critical points remain unresolved and currently preclude a clear assessment of the hardware contribution and system-level viability.

Comment#1

While the HfO₂ gate lowers the threshold for read/display operation, the conductance programming for in-memory computing still relies on high amplitude pulses (−30V to −35V), leaving open questions about true low-power operation. The required programming voltages also raise concerns about device lifetime. Providing endurance data under these conditions (e.g., cycles-to-failure or stability over 10³ - 10⁴ cycles) would help assess reliability.

Comment#2

Fig. 4b and Fig. S17 do not report the essential write/read pulse parameters (amplitude, width, inter-pulse interval, and related settings), leaving the methodology unclear and preventing meaningful energy or latency estimates.

Comment#3

Please define “post-training” and “post-development” explicitly in both the main text and the Fig. 5f caption and clarify whether the weights were obtained ex-situ (software training/simulation) and subsequently programmed into hardware. Also, since the input vector is 25×1 (from a 5×5 image) whereas the weight map is shown as 5×5, a short explanation of how the 25×1 input corresponds to the 5×5 weight visualization would improve interpretability. (In Fig. 5f)

Comment#4

The manuscript does not clearly describe the neural network architecture (layers, activations, parameter count), the training dataset (type/size) and split, or software-hardware accuracy comparisons. Without these, reproducibility is limited and the hardware claim is difficult to interpret.

Reviewer #2

(Remarks to the Author)

I have carefully reviewed the authors' rebuttal and the revised manuscript describing the IPPMLED architecture. The authors have adequately addressed all the major concerns raised in the initial review. In particular, the transition from a 5×5 proof-of-concept to a 16×16 IPPMLED array demonstrates the scalability of the design and is supported by appropriate statistical analysis of device-to-device uniformity. Furthermore, the integration of high-κ HfO₂ dielectrics effectively reduces the driving gate voltages to a more practical range of -4 V to +4 V, successfully addressing the concern regarding high operating voltage and energy efficiency.

The revised manuscript also provides thorough technical validation for the system's high-speed operational capabilities. By exploiting the short-term plasticity of the synaptic transistors, the authors demonstrate stable image refreshing at standard rates of 60 Hz, 120 Hz, and 240 Hz, which clarify how the device maintains display performance alongside its in-memory computing functions. The inclusion of detailed fabrication flowcharts and comprehensive micro-LED characterization metrics further enhances the technical rigor of the manuscript. Given these substantial improvements and the successful experimental validation of the proposed architecture, I believe the revised manuscript now meets the standards for publication in Nature communications.

Reviewer #3

(Remarks to the Author)

Version 2:

Reviewer comments:

Reviewer #1

(Remarks to the Author)

The second revision provides the additional data and clarifications needed to address the outstanding concerns from earlier rounds. The organization is improved, figure annotations are clearer, and the new methodological details allow a fair assessment of the hardware contribution and system feasibility. With the current revisions, the manuscript is suitable for publication in Nature Communications.

Response to Reviewer #1:

Reviewer #1 (Remarks to the Author):

This study introduces an in-pixel processed micro-LED display (IPPMLED) architecture that integrates MoS₂-based synaptic transistors with micro-LEDs for combined pixel driving and memory-related functionality. The system demonstrates luminance levels above 3×10^5 cd/m², integration density of 14,000 units/cm², and operation up to 5000 Hz, with reported cycling stability. Using voltage-luminance modulation and multilevel conductance characteristics, the device is applied to image enhancement and MCU-assisted deblurring. A proof-of-concept 5×5 matrix is presented, showing noise suppression and image reconstruction, with recognition accuracy on the Fashion-MNIST dataset improving from 83.79% to 96.31%. While these results suggest the potential of combining display and processing functions within individual pixels, the demonstrations are limited in scale, and further work is needed to verify scalability, power efficiency, and stability in practical contexts. The concept is of interest, but revisions are necessary to clarify the scope of the hardware contribution and address methodological issues. The detailed comments are as follows:

The authors deeply thank Reviewer #1 very much for carefully reviewing our manuscript and providing us with professional and constructive comments, which have been valuable in enhancing the quality and stringency of our work. Also, we sincerely appreciate that such positive feedback on its significance can be obtained from Reviewer #1. In response to Reviewer #1's suggestions, additional array fabrication experiments, rigorous thinking of the hardware contribution, systematic literature survey, practical hardware implementation, and corresponding modifications have been made, including: **(1) Exploration of device scaling pathways and supplemental experiments for fabricating larger IPPMLED array (16×16)**, achieved through the optimization of large-area material transfer and micro-LED bonding processes (**Figure R1**). **(2) Reduction of operating voltage and power consumption via dielectric layer engineering**, coupled with a systematic literature review to benchmark the power efficiency of the IPPMLED against current display systems (**Table R1**). **(3) Reorganization of the IPPMLED-based display system architecture and revision of the corresponding figures**, providing a clearer delineation of processing roles and a refined balance between in-pixel and off-chip computing (**Figure R4** and **Figure R8**). **(4) Hardware implementation of image display enhancement and neural network deployment** to solidify the hardware contribution of the work (**Figure R3, Figure R5, Figure R6, and Figure R7**). All revisions have been carefully incorporated into the manuscript, with detailed point-by-point responses to Reviewer #1's comments provided in the revised version, where corresponding modifications are clearly highlighted.

1. The devices operate at relatively high gate voltages due to the use of a 300 nm SiO₂ dielectric, which raises concerns regarding energy efficiency for the portable applications (e.g., AR/VR systems). Although the supplementary information briefly mentions HfO₂ as a possible replacement to lower the operating voltage, the manuscript

does not provide a systematic analysis of overall power consumption. For targeting edge computing applications, such quantitative evaluation is essential to support the claimed applicability.

Response:

The authors sincerely appreciate Reviewer #1's critical point regarding the operating voltage and energy efficiency. We fully agree that a systematic and quantitative power analysis is essential for evaluating the applicability of our technology to edge-computing applications. Below, we have conducted a systematic power analysis and literature review for both the vdW driving transistor and the integrated IPPMLED device, with a focus on demonstrating the feasibility of employing a high- κ HfO₂ dielectric to significantly lower the operating voltage and substantially reduce power consumption.

(1) Dielectric material selection and introduction of high- κ HfO₂

In this initial proof-of-concept stage, our primary goal was to validate the novel in-pixel processing architecture. Therefore, we employed a 300 nm SiO₂ dielectric because it provides excellent color contrast for transferred van der Waals materials, ensuring high processing reliability. However, recognizing the need for lower operating voltages in portable and edge-computing systems, we have further developed and integrated a high- κ HfO₂ dielectric process to optimize the operating voltage and power consumption of the device, thereby supporting subsequent systematic integration and application validation. Following the reviewer's suggestion, we optimized key steps—including material transfer, channel patterning, and thermal compression bonding—and successfully fabricated a 16×16 IPPMLED array using HfO₂ as the gate dielectric (**Figure R1a-c**). Also, each pixel is equipped with individually addressable data and gate lines, enabling precise control over its input and output ports. As shown in **Figure R1d-e**, the array exhibits excellent uniformity ($C_v \approx 9.9\%$), and the threshold voltage (V_{th}) is significantly reduced to -0.22 V, confirming the feasibility of our architecture for low-power operation.

(2) Intrinsic energy-efficiency advantages in vdW synaptic transistors

Besides the dielectric engineering, this work employs a synaptic driving transistor with a two-dimensional channel, where the combination of a scattering-suppressing vdW interface and atomically precise electrostatic control provides an inherent foundation for energy-efficient synaptic behavior. To quantitatively evaluate the power consumption of the vdW synaptic transistor for neuromorphic computing, we calculated the energy consumption per synaptic event using the formula $E = I_{peak} \times V \times t$, where I_{peak} is the readout source-drain current, V is the readout source-drain voltage, and Δt is the pulse duration. The calculation result shows that the characteristic energy consumption is as low as 0.14 pJ.

To situate the energy efficiency of our vdW transistor within the current research landscape, we conducted a benchmarking study against recently reported devices (**Figure R1f**). The results demonstrate that the use of a high- κ HfO₂ dielectric enables

a distinct reduction in operating voltage and synaptic power consumption compared to other synaptic transistors, providing a clear and effective pathway for advancing the energy efficiency of IPPMLED devices.

Figure R1 Uniformity and energy efficiency of the IPPMLED device utilizing an HfO₂ dielectric layer. a-c. The 16×16 IPPMLED array with an HfO₂ dielectric layer, showing: (a) an overall optical micrograph, (b) a magnified view of the pixel layout, and (c) its uniform light-emission characteristics. d. Transfer characteristic curves of 100 individual HfO₂-based IPPMLED devices at a fixed data voltage of 3 V. e. Statistical distribution of V_{th} from transfer characteristic curves. f. Benchmarking of energy consumption per spike across various emerging synaptic electronic devices with different channel materials^[1-10].

(3) Overall power assessment of the IPPMLED integrated unit

For addressing the reviewer’s point on application-level power, we characterized the IPPMLED integrated unit. The device operates at data voltage (V_{data}) exceeding 3 V, exhibiting a power consumption of 14.4 μW under a current density of 95.1 A/cm^2 . **Table R1** provides a systematic analysis of recent advances in the LED field, including operating voltage and power consumption, demonstrating the comprehensive advantages of our IPPMLED. Note that the limited test conditions presented in the surveyed studies preclude a unified statistical analysis of power consumption. Moreover, the vdW synaptic transistor-driven IPPMLED features programmable non-volatile conductance, providing another key energy-saving advantage for display processing. Once programmed, the conductance state is inherently retained, eliminating the need for power-intensive external components like constant current sources. This approach drastically cuts the power overhead from data movement and peripheral circuitry.

Table R1 The surveyed studies about the power consumption of various micro-LED technologies.

References	Display type	Driver transistor	V_{data}	V_{gate}	Power consumption
Nat. Nanotechnol. 16, 1231-1236 (2021)	Micro-LED	MoS ₂ driver transistor	8 V	8 V	18 μ W (28 A/cm ²)
Nat. Electron. 6, 216-224 (2023).	Micro-LED	Poly-Si transistor	-5 V	-8 V	1345 μ W
Displays 87, 102997 (2025).	Micro-LED	Si-CMOS	4 V	4.5 V	30 μ W
Adv. Mater. 37, 2416015 (2025) .	Micro-LED	Poly-Si transistor	-4 V	-10 V	~190 μ W (31.6 A/cm ²)
Adv. Mater. 37, 2411999 (2025) .	Micro-LED	/	2.8 V	/	120 μ W
Nano Energy 135, 110613 (2025).	Micro-LED	/	6 V	/	361 μ W
Nat. Commun. 16, 9612 (2025).	Micro-LED	/	3.3 V	/	146.8 μ W
Adv. Optical Mater. 13, 2500271 (2025)	Micro-LED	/	4.3 V	/	3.6×10^4 μ W/cm ² (222 A/cm ²)
Nat. Nanotechnol. 17, 500-506 (2022).	Micro-LED	MoS ₂ transistor	8 V	8 V	~150 μ W
Light Sci. Appl. 12, 258 (2023)	Micro-LED	Si-CMOS	3.2 V	5 V	2.16 μ W (0.37 A/cm ²)
This work	Micro-LED	MoS₂ transistor	3 V	3 V	14.4 μW (95.1 A/cm²)

On the other hand, it should be specifically noted that, since our IPPMLED array is utilized at the device level for weight deployment and the implementation of multiply-accumulate (MAC) operations, its system-level total power consumption is significantly influenced by workload-specific pulse patterns and peripheral circuits (such as ADC, MCU, etc.), making it difficult to estimate. This absence of a standardized metric for this novel class of devices is also observed in several previously published studies on display driving (for example, *Nat. Nanotechnol.* 16, 1231–1236 (2021), *Nat. Nanotechnol.* 17, 500–506 (2022)). Nonetheless, we believe that the low-power characteristics demonstrated at the device level have already laid a solid foundation for constructing future low-energy intelligent display systems.

In summary, in response to the reviewer’s comments, we have included a detailed discussion on the dielectric optimization approach, supported by extensive power consumption data and a comparative study with state-of-the-art display studies. These additions collectively serve to substantiate the promising low-power characteristics of our device.

Corresponding changes made in the manuscript:

To address Reviewer #1's concerns, we have supplemented detailed power consumption data and benchmarking analysis in **Figure S16** and **Table S2**, while also elaborating on the dielectric optimization pathway in the **Figure S13** and **Figure S14**. These additions collectively provide a more comprehensive evaluation of the energy efficiency potential of our IPPMLED technology. We sincerely appreciate the reviewer's insightful comments, which have significantly enhanced the stringency and completeness of our work.

“Note that 300 nm SiO₂ is employed as the gate dielectric during the early-stage exploration and validation of the proposed display architecture, owing to its advantages in better color contrast for transferred vdW materials and hence improving the processing reliability.” on Page 4.

“On the other hand, to meet the low-power requirements of edge computing applications, we further investigated the potential of high- κ dielectrics in replacing SiO₂ gate insulator for power consumption optimization (38, 39). **Figure 5a** shows the successfully fabricated 16×16 IPPMLED arrays with an increased pixel density of 336 PPI by reducing the pixel pitch. The detailed process optimization for large-scale array integration is provided in **Figure S13** and **Figure S14**, showcasing refined methods for large-area vdW material transfer and micro-LED bonding techniques. As expected, the IPPMLED device with dielectric engineering exhibits a notable reduction in the driving voltage, while maintaining good device-to-device uniformity in both V_{th} and on/off-state across 100 devices ($C_v=9.9\%$) (**Figure S15**). Correspondingly, the inherently low power consumption of the synaptic transistor translates into a great energy efficiency advantage for the integrated LED device over current display technologies (**Figure S16 and Table S2**).” on Page 10.

2. In Figure 4f, the authors demonstrate potentiation behavior through output current changes under repeated pulses, illustrating a learning-forgetting-relearning cycle. While this result confirms short-term synaptic plasticity, the analysis is limited to tens of seconds. For establishing robust memory functionality, additional data on long-term retention and potentiation–depression characteristics (linearity, symmetry) are necessary.

Response:

We sincerely appreciate the reviewer's insightful comments regarding the long-term retention and conductance modulation characteristics of our device. We agree that long-term potentiation (LTP) and the linearity/symmetry of synaptic weight updates are essential for robust neuromorphic memory functionality. In response, we have conducted additional experiments to thoroughly evaluate these aspects in our IPPMLED devices.

(1) Long-term retention characteristics of our IPPMLED

To validate the LTP of our device, we conducted an extended retention test by applying electrical pulses with higher amplitude (from -30 V to -35 V) and monitoring the conductance state over a prolonged period. As shown in **Figure R2a**, under a protocol of 30 consecutive electrical pulses followed by a 150-second recovery period, the integrated IPPMLED exhibited learning-experience properties characterized by a sustained enhancement of the EPSC response beyond the original current level. This is evidenced by the progressively intensified optical output shown in the inset.

Figure R2 Long-term synaptic plasticity characteristics of the IPPMLED device. a. Learning-forgetting-relearning behavior of the IPPMLED devices, illustrated by the accumulation of EPSC after 30 consecutive pulses followed by a 150-second delay. **b.** Potentiation-depression characteristics of the IPPMLED device. **c.** Linearity evaluation of the potentiation and depression processes.

(2) Linearity and symmetry of potentiation and depression behaviors

The linearity and symmetry of conductance updates are key metrics for achieving high-accuracy neuromorphic learning. We systematically applied multiple pulse sequences to characterize both the potentiation and depression processes. Based on the established methodology (*Nat. Commun.* **13**, 6431 (2022), *Nat. Commun.* **14**, 6184 (2023)), **Figure R2b-c** summarizes the nonlinearity and symmetry of the weight updates, meeting the requirements for high-performance neuromorphic computing tasks.

These supplementary experiments and data robustly demonstrate that our IPPMLED device exhibits key synaptic characteristics—long-term retention and controllable conductance updates that align with the demands of advanced neuromorphic systems. We sincerely thank the reviewer for these valuable suggestions, which have significantly improved the completeness and persuasiveness of our study.

Corresponding changes made in the manuscript:

In response to the reviewer’s suggestion, we have performed additional experiments to characterize the device's long-term retention properties, which are now included in an updated **Figure 4f**. Furthermore, we have supplemented the analysis with the potentiation–depression characteristics in **Figure S17**, based on which the linearity and symmetry of the conductance updates were quantified.

“Figure 4f demonstrates the learning–experience characteristics of the integrated IPPMLED cell under a stimulation protocol of 30 consecutive electrical pulses followed by a 150-second recovery period. Specifically, the EPSC response is shown

to increase, surpassing the previous baseline levels, as evidenced by the progressively intensifying light output in the inset image. In summary, the IPPMLED device exhibits diverse conductance plasticity characteristics under different pulse conditions, enabling versatile display operating modes.” on Page 9.

“Furthermore, the device exhibits controllable conductance modulation, akin to the flexible synaptic plasticity, under different pulse programming conditions (Figures S17b-e). Specifically, the conductance state demonstrates non-volatile conductance retention lasting over 150 seconds with wider-pulse stimulus, highlighting its potential for neural network weight deployment applications.” on Page 10.

3. A central claim of the manuscript is that the device enables hardware-level contrast stretching by exploiting its piecewise voltage-luminance response. However, the detailed explanation of this process, including the mapping relationship and supporting histograms, is presented only in the Supplementary Information. Given its importance to the overall contribution, this description should be included in the main text to substantiate the in-situ processing claims. Otherwise, the role of hardware in the contrast stretching demonstration remains unclear.

Response:

We sincerely appreciate the reviewer’s insightful comments for highlighting the importance of the hardware-level contrast stretching demonstration. We agree that a detailed explanation of the mapping relationship and the underlying mechanism is essential to substantiate our claim of in-situ display processing. In the revised manuscript, we have revised the corresponding descriptions and reorganized the main text and the corresponding Supplementary Information to the main text, with the following key clarifications:

(1) Correspondence between the voltage–luminance response and the contrast stretching algorithm

As we obtained in Figure 3b in the manuscript, the proposed IPPMLED device exhibits a distinct piecewise voltage–luminance transfer characteristic: a non-emissive ‘dead zone’ at low voltages, an approximately linear response region at intermediate voltages, and a saturation regime at high voltages. This behavior closely aligns with the mathematical principle of conventional contrast stretching algorithms, which compress the low- and high-gray-level distributions while stretching the mid-tones. Such an operation can be represented by a piecewise linear transformation as follows:

$$P_{out} = \begin{cases} 0, & P_{in} < P_{min} \\ \alpha(P_{in} - \mu) + \mu, & P_{min} < P_{in} < P_{max} \\ 255, & P_{in} > P_{max} \end{cases} \quad (1)$$

where P_{in} and P_{out} are the input pixel value and output pixel value, respectively, and α is the contrast adjustment factor ($\alpha > 1$ enhances contrast, $0 < \alpha < 1$ reduces it). The term μ refers to the mean pixel intensity of the entire image. The threshold P_{min} defines the weak-interference pixel range, where values of P_{in} below P_{min} are mapped to 0 in the output. Similarly, P_{max} marks the strong-feature pixel threshold, where values

of P_{in} above P_{max} are stretched to the maximum output value of 255. This process physically widens narrow-band pixel distributions via pixel remapping.

Based on the well-characterized piecewise voltage-luminance response of our IPPMLED, we implemented device-level piecewise linear pixel remapping via voltage-modulated luminance (Equation 2):

$$B(V) = \begin{cases} 0, & V < V_{min} \\ \beta V_{in} + \gamma, & V_{min} < V < V_{max} \\ 2 \times 10^4, & V > V_{max} \end{cases} \quad (2)$$

where V_{in} is the regulation voltage, β is the regulation factor, and γ is the regulation constant.

Figure R3 Contrast-stretching image processing based on the IPPMLED. **a.** Relationship between the voltage-luminance curve of the IPPMLED and the mapping function of the nonlinear contrast-stretching algorithm. **b.** Photograph of the wire-bonded IPPMLED array on a printed circuit board, along with an optical micrograph of the 5×5 device die (scale bar: $50 \mu m$). **c.** Processing schematic of voltage encoding of input images, real-time image enhancement using the IPPMLED array, and optical display of the enhanced results. **d.** Operational schematic detailing the contrast-stretching process implemented by the IPPMLED.

e. Input images of characters “S” with artificially added noise (i), the programming voltage (ii), the corresponding luminance outputs (iii), and the resulting enhanced images produced by the IPPMLED array (iv).

As illustrated in **Figure R3a**, the input–output response curve of the algorithm corresponds well with the measured voltage–luminance behavior of the IPPMLED: low-gray (low-voltage) regions are compressed, high-gray (high-voltage) regions approach saturation, and intermediate-gray regions exhibit near-linear amplification. This inherent nonlinear mapping capability enables the direct implementation of contrast stretching at the hardware level, without requiring additional digital circuits or software processing.

(2) Mapping the relationship between input pixel values and driving voltages

To experimentally validate this concept, we fabricated several 5×5 device dies and demonstrated the contrast stretching operation, where each IPPMLED pixel is individually addressable through dedicated data and gate lines for precise control of both input programming and optical output (**Figure R3b**). As shown in **Figure R3c-d**, based on the piecewise voltage–luminance response, input image gray levels were converted into driving voltages via a predefined piecewise scaling method, with low-gray pixels being mapped into the dead-zone region for compression, high-gray pixels driven into the saturation regime for suppression, while mid-gray pixels were assigned to the linear region where they undergo effective amplification. Through this voltage–gray-level mapping, the distribution of pixel values is physically reconfigured, achieving contrast enhancement effects comparable to software algorithms (**Figure R3e**).

Corresponding changes made in the manuscript:

In response to the reviewer’s comment, we have provided a clearer description of the IPPMLED-based contrast stretching workflow. Besides providing the specific contrast stretching formulation and pixel-to-voltage mapping-based experimental process in **Note S2** and **Figure S21**, we have also updated the conceptual diagram of the contrast stretching algorithm and its mapping scheme onto the hardware, which are now presented in **Figure 5b** to make a clear explanation. We sincerely appreciate the reviewer’s insightful comments, which have greatly improved the readability and stringency of our manuscript.

“As illustrated in **Figure 1a**, the IPPMLED architecture unifies the dual capabilities of in-situ image enhancement and on-chip weight updating for image recognition under a unified hardware platform. Specifically, by leveraging its gate-tunable luminescence properties, the device enables real-time contrast stretching (**Figure 1a(i)**). This process dynamically remaps input pixel distributions for instantaneous display.” on Page 4.

“As shown in **Figure 5b**, the input image pixels are first converted into driving voltages according to the device-specific voltage–luminance characteristic, which intrinsically aligns with the contrast-stretching algorithm. In this scheme, low pixel values are

mapped to the sub-threshold voltage region (no emission), high values to the saturation region, and intermediate values to the linearly stretched operating region, respectively, thereby enabling immediate contrast-enhanced display.” on Page 11.

“Specifically, the pixel values of the input image are converted to corresponding voltage signals spanning from -4 V to 4 V and injected into the IPPMLED array. The detailed pixel-to-voltage mapping during the contrast stretching procedure is provided in **Note S2** and **Figure S21**. Each pixel element autonomously processes its input voltage based on intrinsic voltage-dependent luminance characteristics, providing real-time optical feedback. Distinctly, with the device-specific contrast stretching, the input images are enhanced by the IPPMLED array that suppresses noise pixels and boosts feature pixels, effectively yielding clear images and operating steadily (Figure 5d).” on Page 12.

“Based on the voltage–luminance modulation properties and non-volatile conductance updating characteristics of the driver transistors, a hardware platform centered around the IPPMLED array has been constructed to achieve integrated in-situ image processing and real-time display. Specifically, the fabricated IPPMLED array demonstrates image enhancement of noisy inputs (“S”, “H”, and “U”) through piecewise linear voltage–luminance characteristics that closely match pixel remapping schemes.” on Page 13.

4. The description of the deblurring demonstration lacks sufficient detail. Although the manuscript states that the process is “MCU-assisted,” it does not clearly explain which parts are carried out by the device and which by the MCU. Moreover, the algorithmic details are missing. It is not specified whether a neural network was used, what network architecture or parameters were adopted, or how the computation was divided between the MCU and the device. Without this information, it is difficult to determine whether the demonstration truly reflects in-pixel processing or is largely dependent on external computation. A clearer and more rigorous explanation of the deblurring methodology is therefore essential.

Response:

We sincerely thank the reviewer for the professional comments regarding the description of the ‘MCU-assisted deblurring’ demonstration, which have helped us improve the completeness of the hardware demonstration section. In this work, we explored image processing capabilities based on the proposed IPPMLED design, including image enhancement and recognition tasks, to demonstrate its potential for image deblurring applications. To summarize, both image enhancement and recognition form integral parts of our display optimization workflow. **Specifically, the in-memory computing acceleration for image recognition is performed in situ by the IPPMLED array. In this process, the MCU assists with functions such as decision support, image encoding/decoding, and triggering advanced deblurring procedures with the aid of a computer under specific conditions.** We apologize for the lack of a detailed description regarding the hardware demonstration in the original manuscript. Following the reviewer’s suggestion, we have now provided comprehensive clarifications and additions in the revised manuscript regarding the

architecture of the in-pixel processing system, the hardware workflow, and algorithmic details to better illustrate the potential of our device. Below, we provide a detailed supplementary explanation:

(1) Roles and responsibilities in the IPPMLED system

As mentioned earlier, we explored display processing demonstrations in this work through image enhancement, image recognition, display decision-making, and display calibration to achieve display optimization. Leveraging the in-memory computing acceleration capability of IPPMLED, low-level image enhancement and certain high-level image processing workflows (such as array-based neural network image recognition) are performed within the IPPMLED array. The MCU is responsible for auxiliary functions such as data buffering and image encoding/decoding. **Figure R4** provides a schematic diagram of the specific image processing workflow. As shown, following in-situ contrast stretching, the IPPMLED array serves as the computational hardware for neural network-based recognition. It performs batched weight deployment and in-situ MAC operations, which collectively represent the most energy-demanding and central part of the inference process. Based on the recognition results, the system then determines whether to trigger MCU-assisted image deblurring. Specifically, when recognition accuracy falls below a predefined threshold, the MCU activates an advanced deblurring procedure. The refined image is then re-encoded and fed back to the IPPMLED for display and another recognition cycle until high-accuracy identification is achieved. Conversely, if the recognition accuracy meets or exceeds the threshold, the system bypasses high-level processing to prevent unnecessary computational overhead.

Figure R4 Schematic illustration of the contrast stretching and image recognition flow charts of the proposed in-pixel image processing architecture.

(2) IPPMLED array-based image enhancement and neural network image recognition

Firstly, the IPPMLED device exhibits a piecewise linear voltage–luminance transfer characteristic, comprising three distinct regions: a non-emissive region at low gate voltages, a quasi-linear modulation region at intermediate voltages, and a saturation region at high voltages, where luminance can be precisely and predictably controlled.

This characteristic aligns well with the contrast-stretching algorithm, which remaps input pixel values to produce an output with a redistributed and enhanced pixel value distribution. As shown in **Figure R5a**, we fabricated several 16×16 pixel arrays to validate the display processing capability. Specifically, a 5×5 device die is employed, in which each pixel is individually addressable via dedicated data and gate lines, enabling precise control of input programming and optical output. Based on the voltage–luminance characteristics, input image pixel values are mapped to specific voltage ranges. The IPPMLED physically responds differently to pixels in different voltage regions (non-emissive, linear emission, or saturated emission), thereby displaying the stretched image directly in the hardware. It should be emphasized that during the contrast-stretching process, external devices (such as the MCU (STM32F103RCT6)) only provide the initial voltage signals and do not participate in any computation related to luminance output.

Figure R5 Demonstration of the in-pixel display processing based on IPPMLED arrays.
a. Photograph of the display processing platform, showing the matrix-configured IPPMLED array mounted on a custom PCB for multi-channel measurement. **b.** The weight distributions of an IPPMLED array from both post-training simulation and post-deployment. **c.** Current

recordings from the IPPMLED array while executing MAC operations. **d.** Confusion matrices for the recognition of IPPMLED-enhanced images (i) versus original images (ii). **e.** Flowchart of the weight mapping and inference process, incorporating the IPPMLED array, peripheral electronics.

Secondly, leveraging the in-memory computing acceleration of the IPPMLED array, we also implemented neural network hardware-based image recognition inference tasks. As shown in **Figure R5b**, pre-trained and quantized neural network weights are quantized and deployed in the form of IPPMLED device conductance. When the input voltage vector—encoded from the enhanced image—is applied to the array, the MAC operations between the input voltage vector and the weight matrix are performed in parallel within the array, in accordance with Kirchhoff's current law (**Figure R5c**). This constitutes the most energy-intensive and core computational part of neural network inference. Simultaneously, the input voltages directly drive the IPPMLED to achieve in-situ image display. During the image recognition process, since image data are written row by row, the MCU is responsible for storing intermediate row results, scheduling timing, and reading output currents, but it does not perform large-scale MAC operations. **Figure R5d** presents an inference result with a recognition accuracy of 99.29%, demonstrating the compatibility of the IPPMLED with peripheral circuits like the MCU and its suitability for system-level applications.

(3) MCU- and computer-assisted high-level image deblurring

In the high-level image deblurring demonstration, due to the limited scale of the IPPMLED array, it is challenging to fully map all computational steps (such as display decision-making and calibration) onto the array. Therefore, certain deblurring-related computations, including display decision-making and calibration, are executed in the computer and MCU (STM32F103RCT6) (**Figure R5e**). In this process, the MCU re-encodes the deblurred image data into corresponding voltage sequences and inputs them again to the array for secondary image enhancement, display, and a second round of recognition, entering a new cycle of accuracy-based decision-making. Although this high-level processing relies on the MCU and computer, the workflow amply demonstrates the inherent compatibility of the proposed IPPMLED in-memory computing array with standard digital processors like the MCU, confirming the feasibility and flexibility of this architecture.

In summary, we reiterate that in the image deblurring tasks demonstrated, key functions such as neural network weight deployment, analog-domain MAC operations, and image display are implemented by the proposed IPPMLED array, while certain tasks such as image encoding/decoding and hierarchical decision-making are executed by standard signal processing components like the MCU. This architecture effectively combines the high energy efficiency and low latency of in-memory computing with the flexibility and precision of digital processing. Furthermore, we emphasize that the main contribution and value of this work lie in proposing a van der Waals transistor-driven micro-LED pixel unit and demonstrating in-pixel in-memory computing, thereby significantly reducing the power consumption and latency

associated with massive data transfer in display driving, and providing an efficient paradigm for future intelligent edge displays. In addition, based on the reviewer's valuable suggestions, we will further pursue the design and fabrication of higher-integration display-driving computing arrays in the future to achieve more intelligent, high-resolution micro-display systems.

Corresponding changes made in the manuscript:

To address the reviewer's comment, we have updated the **main text** to unambiguously clarify the distinct roles and collaboration between the IPPMLED and the MCU. This clarification is visually supported by the updated conceptual diagram in **Figure 5b**. Additionally, the experimental test platform and corresponding flowchart have been included as **Figure S19**. The neural network architecture and specific parameters are documented in the **Method** section. We believe these revisions have substantially improved the manuscript, and we thank the reviewer for their input, which was instrumental in achieving this greater level of clarity and systematic stringency.

“In this work, the IPPMLED features gate-tunable luminance response and in-memory computing capability, offering significant advantages for in-situ display processing. To fully harness the advantages of the proposed devices, both contrast stretching and image recognition are taken as classic examples to exhibit how the IPPMLED array provides a processing platform towards an optimal balance between in-pixel processing and off-chip management.” on Page 11.

“To comprehensively assess the capability of the proposed IPPMLED in reshaping the intelligent in-situ display landscape, a synchronized display processing pipeline capable of handling representative multi-level tasks has been designed. As shown in **Figure 5b**, the input image pixels are first converted into driving voltages according to the device-specific voltage–luminance characteristic, which intrinsically aligns with the contrast-stretching algorithm. In this scheme, low pixel values are mapped to the sub-threshold voltage region (no emission), high values to the saturation region, and intermediate values to the linearly stretched operating region, respectively, thereby enabling immediate contrast-enhanced display. To further optimize the display output, an in-pixel recognition function is proposed. Specifically, the IPPMLED array, serving as the neural network accelerator, enables efficient hardware inference by executing batched weight deployment and in-situ MAC operations. The system subsequently determines whether to trigger MCU-assisted image deblurring according to the obtained recognition results. For instance, if the obtained recognition accuracy falls below a predefined threshold, an advanced deblurring procedure would be activated with the assistance of the MCU (STM32F103RCT6) and back-end processors. Conversely, a higher recognition accuracy would bypass high-level processing to achieve the high-quality display.” on Page 12.

“The quantitative assessment is further considered to validate the contrast stretching operation, providing better guidance for subsequent refinement of the display architecture. We demonstrate an in-pixel image recognition using an IPPMLED-based feedforward neural network, applied to both original noisy letter images and their

IPPMLD-enhanced counterparts (Figure 5e). As shown in Figure 5f, the trained weights are extracted and deployed onto the IPPMLD array via non-volatile conductive retention, achieving a complete correspondence with the target weight matrix. Subsequently, the letter images (5×5 pixels) are transformed into 25×1 vectors to serve as inputs, where parallel MAC operations are performed with the deployed weights, enabling highly parallel, low-power analog computing and in-situ execution of the inference process within the IPPMLD. Figure 5g illustrates a representative current summation process across the array during this operation. The resulting MAC values are temporarily stored in the readout circuit and buffer registers, after which weight refreshing and array reuse enable the processing of subsequent batches, ultimately yielding the recognition result.” on Page 12.

“We develop a multilayer perceptron (MLP) model for character classification on a custom dataset of 5×5 grayscale letter images. The dataset contains 26 classes (A–Z) with 1000 samples per class, totaling 26000 images. Each character is represented as a 5×5 grayscale array and is augmented with controlled-amplitude random noise to improve robustness to image perturbations. All samples undergo normalization before being fed into the network. The model consists of a three-layer fully connected architecture: an input layer, two hidden layers, and an output layer. Each hidden layer is followed by a ReLU activation function to enhance nonlinear representation. The network is trained for 250 epochs using the AdamW optimizer, minimizing the cross-entropy loss for classification. The dataset is split into separate training and validation sets, with validation accuracy recorded after each epoch to monitor generalization performance.” in the Method.

“To achieve hardware-accelerated neural network inference, an integrated system integrating an IPPMLD array, peripheral electronics (INA226A ADC, DAC7311 DAC), and a microcontroller unit (MCU, STM32F103RCT6) is developed. The IPPMLD array is configured in a matrix layout and wire-bonded onto a PCB (ASM Pacific Technology, IHAWK AERO). Programming voltages are supplied by an NI-PXI 4163 modular instrumentation system, integrated with a 24-channel PXI-2532B multiplexer for row and column addressing. During inference, the trained synaptic weights are deployed onto the IPPMLD array via non-volatile conductance modulation. Then, the input image—a 5×5 pixel grayscale frame—is encoded into programmable voltage signals proportional to pixel intensity and applied to the array for in-situ MAC operations. The resulting output currents are then amplified and converted by peripheral readout circuitry, with the processed signals being temporarily stored in buffer registers. Following this, the array undergoes weight refreshing and is reused to process subsequent image batches, ultimately generating the final recognition result.” in the Method.

5. Across the demonstrations, it is unclear whether true in-situ processing was actually carried out at the hardware level using MoS₂ synaptic transistors. The descriptions of contrast stretching and MCU-assisted deblurring do not provide sufficient evidence that these functions were intrinsically realized within the device rather than externally emulated. For the central claim to be convincing, the authors should present a more

precise explanation of how in-situ processing was implemented, along with clear experimental evidence demonstrating that the computation indeed occurred in the hardware.

Response:

We sincerely thank the reviewer for the thorough analysis and constructive feedback on our manuscript. We fully understand and acknowledge the concern regarding the need to more clearly distinguish between “in-situ processing within the device” and “MCU-assisted processing”. In response, we have provided more experimental details and results to demonstrate that both contrast stretching and matrix MAC operations are indeed intrinsically implemented at the hardware level, relying on the inherent physical properties of the MoS₂ synaptic transistor array and the aid of the MCU/CPU. Meanwhile, the image encoding/decoding and display decision-making in the deblurring workflow are dominated by the MCU and CPU modules.

Figure R6 Demonstration of the in-pixel image enhancement based on IPPMLED arrays. **a.** Processing schematic of voltage encoding of input images, real-time image enhancement using the IPPMLED array, and optical display of the enhanced results. **b.** The diagram illustrates the workflow from a noisy input image to the enhanced display output via the IPPMLED. **c.** Input images of characters “S”, “H”, and “U” with artificially added noise (i) and the resulting enhanced images produced by the IPPMLED array (iii). **d.** Photograph of the implemented image enhancement and weight deployment platform, showing the matrix-configured IPPMLED array mounted on a custom PCB for multi-channel measurement.

(1) In-situ implementation of contrast stretching

As we mentioned above, the contrast stretching function is fully achieved by exploiting the intrinsic nonlinear electro-optical transfer characteristic of the micro-LED device itself, without any involvement of external computational units. By applying input voltages directly to each micro-LED pixel, the output luminance is determined solely by the device's inherent input voltage–output luminance response curve. We pre-characterized this curve, which exhibits three distinct piecewise-linear regions: a dead zone at low voltages, a quasi-linear modulation region at intermediate voltages, and a saturation region at high voltages. This transfer characteristic naturally aligns with the mathematical operation of contrast stretching, which remaps input pixel values to enhance mid-tone distribution, as illustrated in **Figure R6a** and **Figure R6b**.

To experimentally validate this, we fabricated multiple 5×5 device dies, where each pixel is individually addressable via dedicated data and gate lines, enabling precise control of input programming and optical output. By mapping the input image pixel data to specific voltage ranges, the micro-LED physically responds differently to pixels in different voltage regions (non-emissive, linear emission, or saturated emission), thereby directly displaying the stretched image through its inherent physical behavior (**Figure R6c**). Throughout this process, external devices such as the MCU only supplied the initial voltage signals and were not involved in any computation related to luminance output.

(2) In-situ implementation of matrix multiplication-accumulation for computing-in-memory

Recognizing the limited scope of contrast stretching for comprehensive image processing, we further extended the system to support further deblurring tasks. This process is divided into array-based neural network inference and MCU-dependent algorithmic processing.

In the array-based neural network recognition process, pre-trained and quantized neural network weights are mapped and programmed into the IPPMLED array in the form of device conductance states. **Figure R6d** presents the experimental test platform employed in this study. The supplemental experimental test setup and the weight programming results are displayed in **Figures R7**, respectively, showcasing the key experimental conditions and data obtained. Specifically, when the input voltage vector—encoded from the enhanced image—is applied to the array, the MAC operations between the input voltage vector and the weight matrix are performed in parallel within the array, in accordance with Kirchhoff's current law. This constitutes the most energy-intensive and core computational part of neural network inference. Simultaneously, the input voltages directly drive the micro-LED to achieve in-situ image display. During this process, the MCU (STM32F103RCT6) is only responsible for auxiliary functions, including writing parameters, scheduling timing, and reading output currents, but does not perform large-scale MAC operations.

When the recognition accuracy falls below a predefined threshold, more complex deblurring procedures are triggered. Owing to the current limitations in array scale, it is not yet feasible to fully map the entire deblurring algorithm onto the array. Therefore,

these advanced deblurring computations are executed within the computer and the MCU. The MCU (STM32F103RCT6) re-encodes the deblurred image data into corresponding voltage sequences and feeds them back into the array for display. The deblurred image then re-enters the recognition cycle until high-precision recognition is achieved, during which the IPPMLED array performs luminance calibration and outputs the final deblurred image.

Figure R7 Hardware inference for image recognition based on an IPPMLED array. a. Weight deployment and hardware inference process. The heatmaps show the distribution of conductance states assigned to the hardware array in respect to the trained weight distribution. **b.** The column current measurements of an IPPMLED array during the MAC process. **c.** Confusion matrix showing the recognition results for the hardware-based inference performed on the IPPMLED array.

In summary, we have clearly delineated the respective roles: core in-situ processing such as contrast stretching and MAC operations, is realized by the intrinsic physical properties of the IPPMLED array, while tasks requiring higher programmability and complex decision-making are handled by the MCU and computer. We believe these clarifications and supplementary data convincingly demonstrate the in-situ processing capabilities of our MoS₂-based synaptic transistor array at the hardware level.

Corresponding changes made in the manuscript:

To clarify the IPPMLED's in-situ functions in contrast stretching and image recognition and its collaboration with the MCU, we have updated the main text and revised the

conceptual diagrams in **Figure 1a** and **Figure 5b**. The specific experimental procedures and system structure have also been detailed in **Figures 5f-h** and **Figure S19**.

“By exploiting the pixel-level weight programmability of the IPPMLEDs, the trained neural network weights for image recognition can be deployed directly onto the IPPMLED arrays for hardware-accelerated inference. Consequently, the in-situ processed images achieve a recognition accuracy of 99.29%, significantly outperforming unprocessed inputs (79.81%).” on Page 3.

“In this work, the IPPMLED features gate-tunable luminance response and in-memory computing capability, offering significant advantages for in-situ display processing. To fully harness the advantages of the proposed devices, both contrast stretching and image recognition are taken as classic examples to exhibit how the IPPMLED array provides a processing platform towards an optimal balance between in-pixel processing and off-chip management.” on Page 11.

“To comprehensively assess the capability of the proposed IPPMLED in reshaping the intelligent in-situ display landscape, a synchronized display processing pipeline capable of handling representative multi-level tasks has been designed. As shown in **Figure 5b**, the input image pixels are first converted into driving voltages according to the device-specific voltage–luminance characteristic, which intrinsically aligns with the contrast-stretching algorithm. In this scheme, low pixel values are mapped to the sub-threshold voltage region (no emission), high values to the saturation region, and intermediate values to the linearly stretched operating region, respectively, thereby enabling immediate contrast-enhanced display. To further optimize the display output, an in-pixel recognition function is proposed. Specifically, the IPPMLED array, serving as the neural network accelerator, enables efficient hardware inference by executing batched weight deployment and in-situ MAC operations. The system subsequently determines whether to trigger MCU-assisted image deblurring according to the obtained recognition results. For instance, if the obtained recognition accuracy falls below a predefined threshold, an advanced deblurring procedure would be activated with the assistance of the MCU (STM32F103RCT6) and back-end processors. Conversely, a higher recognition accuracy would bypass high-level processing to achieve the high-quality display.” on Page 12.

“Specifically, the pixel values of the input image are converted to corresponding voltage signals spanning from -4 V to 4 V and injected into the IPPMLED array. The detailed pixel-to-voltage mapping during the contrast stretching procedure is provided in **Note S2** and **Figure S21**. Each pixel element autonomously processes its input voltage based on intrinsic voltage-dependent luminance characteristics, providing real-time optical feedback. Distinctly, with the device-specific contrast stretching, the input images are enhanced by the IPPMLED array that suppresses noise pixels and boosts feature pixels, effectively yielding clear images and operating steadily with negligible optical. (Figure 5d).” on Page 12.

“The quantitative assessment is further considered to validate the contrast stretching operation, providing better guidance for subsequent refinement of the display

architecture. We demonstrate an in-pixel image recognition using an IPPMLED-based feedforward neural network, applied to both original noisy letter images and their IPPMLED-enhanced counterparts (Figure 5e). As shown in Figure 5f, the trained weights are extracted and deployed onto the IPPMLED array via non-volatile conductive retention, achieving a complete correspondence with the target weight matrix. Subsequently, the letter images (5×5 pixels) are transformed into 25×1 vectors to serve as inputs, where parallel MAC operations are performed with the deployed weights, enabling highly parallel, low-power analog computing and in-situ execution of the inference process within the IPPMLED. Figure 5g illustrates a representative current summation process across the array during this operation. The resulting MAC values are temporarily stored in the readout circuit and buffer registers, after which weight refreshing and array reuse enable the processing of subsequent batches, ultimately yielding the recognition result. As expected, the IPPMLED-enhanced images demonstrated a high inference accuracy of 99.29%, outperforming the unprocessed images (79.81%) (Figure 5h and **Figure S22**), enabling high-quality display output while eliminating the need for further processing.” on Page 12.

6. Figure 1a is described simply as a “schematic diagram of the IPPMLED architectures and functional implementation,” but the caption lacks sufficient detail to help readers interpret the figure. The right-hand side shows an image processing part that is only described as “functional implementation” in the caption. However, it is unclear how image processing operations are mapped onto the pixel-level hardware. Also, the label “ADMLED” appears, but this abbreviation is never defined in the manuscript. Since this schematic diagram is crucial to illustrate the computational aspect of architecture, the authors should provide a clearer explanation of the processing pathway and its correspondence with the device functions.

Response:

We sincerely thank the reviewer for this insightful comment regarding Figure 1a. We acknowledge that the original schematic and its caption were indeed too brief in describing the functional implementation, failing to systematically illustrate the correspondence between the image processing flow and the device-level functional modules. Following the reviewer’s suggestion, we have redesigned **Figure 1** in the revised manuscript and updated the caption with more comprehensive details to help readers better understand the system architecture and operational mechanism.

In the new schematic, we have clearly and directly mapped the image processing capabilities of the IPPMLED system to the corresponding device-level hardware functionalities (**Figure R8**). Specifically, the left part of the figure illustrates the concept of the IPPMLED integrated architecture, highlighting its in-memory computing and display integration. The right part clearly delineates two core pathways for in-situ image processing and display in the IPPMLED system:

(1) **Contrast enhancement based on segmented linear voltage–luminance response characteristics of IPPMLED:** Leveraging the inherent segmented voltage–brightness behavior of the IPPMLED array (including the low-voltage dead zone, quasi-linear

modulation region, and high-voltage saturation region), the pixel values of the input image are directly mapped to driving voltages. This enables physical-level contrast stretching and enhanced display without the need for additional digital computation or post-processing.

(2) **Neural network-based image recognition using non-volatile in-memory computing:** By utilizing the programmable and non-volatile properties of synaptic transistors, the trained neural network weights are quantized and stored in the device conductance. When input voltage vectors are applied to the array, the devices perform parallel MAC operations in the analog domain, enabling hardware-accelerated inference and in-situ recognition and display of the input image. Additionally, the MCU serves as an auxiliary unit responsible for input/output codec and operation scheduling. Under conditions of insufficient recognition accuracy or in specific scenarios, it can invoke a computer for secondary image processing. The reprocessed image data are then re-encoded into voltage signals and input to the array for display, thereby forming a closed-loop optimization process.

Figure R8 Schematic diagram of the IPPMLED architectures and functional implementation. The left panel illustrates the integrated memory–computing–display architecture, while the right panel highlights two in-situ image processing pathways: (i) hardware-level contrast enhancement via intrinsic voltage–luminance characteristics, and (ii) in-situ neural network inference by leveraging the non-volatile properties of IPPMLEDs for weight deployment, with the MCU serving as an auxiliary unit for voltage encoding and closed-loop optimization.

Regarding the label “ADMLED”, we apologize for this legacy typographical error from an earlier draft. The correct abbreviation is IPPMLED (in-pixel processing micro-LED). We have corrected this throughout the revised manuscript.

In conclusion, the revisions have systematically clarified the image processing–hardware mapping and underscored our core innovation: a unified vdW transistor-driven micro-LED pixel that merges display with in-memory computing, highlighting its promise for intelligent micro-displays. Our future work will focus on scaling the array integration to push toward systems with higher resolution and intelligence.

Corresponding changes made in the manuscript:

In accordance with the reviewer's suggestion, we have reorganized **Figure 1** and provided a more detailed caption in the revised manuscript to facilitate a clearer understanding of the system architecture and operational mechanism. Additionally, we have also carefully checked all abbreviations in the manuscript to ensure their proper and consistent use.

“As illustrated in **Figure 1a**, the IPPMLED architecture unifies the dual capabilities of in-situ image enhancement and on-chip weight updating for image recognition under a unified hardware platform. Specifically, by leveraging its gate-tunable luminescence properties, the device enables real-time contrast stretching. Simultaneously, the nonvolatile memory characteristics allow persistent retention of synaptic weights, thereby supporting the inference tasks required for robust image recognition within the same pixel array.” on Page 4.

“**a.** Schematic diagram of the IPPMLED architectures and functional implementation. The left panel illustrates the integrated memory–computing–display architecture, while the right panel highlights two in-situ image processing pathways: (i) hardware-level contrast enhancement via intrinsic voltage–luminance characteristics, and (ii) in-situ neural network inference by leveraging the non-volatile properties of IPPMLEDs for weight deployment, with the MCU serving as an auxiliary unit for voltage encoding and closed-loop optimization.” on Page 22

7. In Figure 2e, the average I_{on} is described in the text as $\sim 2.6 \times 10^{-6} A$, but the figure label appears to indicate $I_{on}(x10^6 A)$, which is misleading. Similar typographical errors (e.g., fig.3b, “matrixes” instead of “matrices”) are present. Careful revision is needed to avoid confusion.

Response:

We sincerely thank the reviewer for the careful reading of our manuscript and for identifying these oversights. We apologize for the error in the y-axis label of Figure 2e and other similar typographical and terminology issues. The label in Figure 2e has now been corrected to ‘ $\times 10^{-6} A$ ’. In addition, we have performed a thorough proofreading of the entire manuscript and corrected all such issues, including replacing ‘matrixes’ with ‘matrices’ in Figure 3b. We are grateful for the reviewer's valuable comments, which have helped improve the accuracy and clarity of our article.

Reference:

1. Xu, H., et al. A low-power vertical dual-gate neurotransistor with short-term memory for high energy-efficient neuromorphic computing. *Nat. Commun.* **14**, 6385 (2023).
2. Seo, Seokho, et al. The gate injection-based field-effect synapse transistor with linear conductance update for online training. *Nat. Commun.* **13**, 6431 (2022).
3. Li, L., et al. Floating-gate photosensitive synaptic transistors with tunable functions for neuromorphic computing. *Science China Materials.* **64**, 1219-1229 (2021).

4. Xie, T., et al. Carbon nanotube optoelectronic synapse transistor arrays with ultra-low power consumption for stretchable neuromorphic vision systems. *Adv. Funct. Mater.* **33**, 2303970 (2023).
5. Liang, X., et al. Multimode transistors and neural networks based on ion-dynamic capacitance. *Nat. Electron.* **5**, 859-869 (2022).
6. Ma, M., et al. Multiplexed neurochemical transmission emulated using a dual-excitatory synaptic transistor. *npj 2D Mater. Appl.* **5**, 23 (2021).
7. Li, F., et al. An artificial visual neuron with multiplexed rate and time-to-first-spike coding. *Nat. Commun.* **15**, 3689 (2024).
8. Chen, Y., et al. All two-dimensional integration-type optoelectronic synapse mimicking visual attention mechanism for multi-target recognition. *Adv. Funct. Mater.* **33**, 2209781 (2023).
9. Han, M. J., & Tsukruk, V. V. Trainable bilingual synaptic functions in bio-enabled synaptic transistors. *ACS Nano*, **17**, 18883-18892 (2023).
10. Hao, Z., et al. Retina-inspired self-powered artificial optoelectronic synapses with selective detection in organic asymmetric heterojunctions. *Adv. Sci.* **9**, 2103494 (2022).

Response to Reviewer #2:

Reviewer #2 (Remarks to the Author):

This manuscript presents a novel in-pixel processed display-driven micro-LED display (IPPMLED) architecture that integrates micro-LEDs with molybdenum disulfide (MoS₂) in-memory computing transistors. The goal of this architecture is to overcome the latency and energy consumption bottlenecks of conventional off-pixel processing by co-locating image processing and display at the pixel level. The authors fabricated a 5x5 IPPMLED matrix and demonstrated its capabilities in both low-level and high-level image processing. The reviewer believes that the following revisions will help improve the quality of the manuscript.

The authors deeply thank Reviewer #2 very much for carefully reviewing our manuscript and providing us with positive feedback and constructive suggestions, which have been valuable in enhancing the quality of this work. In response to Reviewer #2's suggestions, additional large-scale array fabrication experiments, systematic literature surveys, display processing mechanisms, practical hardware implementation, and corresponding modifications have been made, including: **(1) Exploration of device scaling pathways and supplemental experiments for fabricating larger IPPMLED array (16×16)**, achieved through the optimization of large-area material transfer and micro-LED bonding processes (**Figure R9-11, R15–R17, Table R2**). **(2) Reduction of operating voltage and power consumption through dielectric layer engineering**, complemented by a systematic literature survey that benchmarks the IPPMLED's power efficiency against current display systems (**Figure R18, Figure R20, Table R4**). **(3) Fabrication and comprehensive optoelectronic characterization of micro-LEDs**, including luminance, current density, and external quantum efficiency (EQE) performance (**Figure R15 and Figure R16**). **(4) Exploration of synaptic plasticity mechanisms underpinning display processing**, clarifying the interplay between long-term retention and rapid refresh capabilities (**Figure R14, Videos R1-6, and Table R3**). **(5) Hardware implementation of image display enhancement and neural network deployment** to solidify the hardware contribution of the work (**Figure R12 and Figure R13**). All suggested revisions have been carefully incorporated into the revised manuscript. A detailed point-by-point response to each comment has been provided, with all corresponding changes clearly highlighted for your convenience.

Comment #1: The authors' claims about "high-resolution display" are ambitious given that the hardware demonstration is limited to a 5 × 5 matrix. Wafer-scale, high-uniformity MoS₂ TFT backplanes remain challenging (many processes still rely on flake transfer or small-area CVD growth), whereas micro-LEDs are already proven at very high pixel densities. This creates a scalability tension between the transistor backplane and the emitter array. While the reported pixel density of 300 PPI is notable, please provide a credible scaling roadmap and demonstrate a larger matrix to validate the claim.

Response:

We sincerely appreciate the reviewer’s insightful comments regarding display resolution and scalability. We fully agree that the claim of ‘high-resolution display’ requires validation through larger array demonstrations. In response, we have provided a more detailed explanation of the scalability roadmap for integrating two-dimensional (2D) material TFT backplanes with micro-LEDs, supplemented with experimental data and analysis from larger arrays to comprehensively assess the scaling potential of our technology.

(1) Strategies for larger array realization and uniformity optimization

As the reviewer correctly pointed out, wafer-scale, high-uniformity MoS₂ TFT backplanes remain technically challenging, leading to a scalability mismatch between the transistor backplane and the micro-LED emitter array. **The primary difficulties arise from the nondestructive transfer of large-area 2D materials, the thermal budget constraints during micro-LED bonding, and the yield and uniformity control of the bonding process.**

To overcome these challenges, based on recent advances in 2D device fabrication techniques, **several improved strategies and roadmaps have been developed to achieve damage-free and highly uniform transfer of large-area 2D materials.** In terms of material transfer, key challenges include regulating the interfacial adhesion between the substrate and the 2D material, as well as optimizing the removal process of the supporting layer. For example, Prof. Zhongfan Liu et al. enhanced the interfacial interaction between graphene and the target substrate by introducing volatile oxygen-containing molecules into the PMMA supporting layer, thereby achieving crack-free and high-quality transfer on a 4-inch wafer scale (*Nat. Commun.* **13**, 4409 (2022)). Prof. Young Hee Lee et al. further reduced polymer residues and minimized structural damage to 2D materials by using anisole as a mild solvent for removing the supporting layer (*Nat. Nanotechnol.* **19**, 34–43 (2024)).

Figure R9 Schematic diagram of the PMMA-assisted wet transfer process for large-area two-dimensional materials. This schematic illustrates the hydrophilic treatment of the substrate and the mild chemical etching process used to remove the PMMA support layer.

In addition, recent studies on micro-LED integration have demonstrated that **low-temperature thermocompression bonding can effectively reduce the thermal budget, enhance bonding quality, and prevent device degradation.** For example,

Hwangbo et al. realized monolithic integration of MoS₂ TFTs with micro-LEDs by maintaining the bonding temperature below 580 °C, thereby avoiding thermal damage to the micro-LEDs (*Nat. Nanotechnol.* **17**, 500–506 (2022)).

Building upon the proposed large-array fabrication and process optimization strategies, and taking into account our laboratory conditions, we carried out supplementary experiments on a larger-scale integrated IPPMLED array. Specifically, we regulated the surface wettability of the target substrate to improve the conformal contact and interfacial uniformity of the transferred 2D films. Meanwhile, a mild chemical etchant was used to remove the PMMA supporting layer, effectively preserving the material integrity and minimizing surface contamination (**Figure R9**). Furthermore, during device integration, we employed a low-melting-point metal as the bonding material to prevent high-temperature degradation and conducted bonding in an inert N₂ atmosphere to further protect the 2D materials (**Figure R10**).

Figure R10 Schematic diagram of the micro-LED low-temperature thermo-compression bonding process. The procedure is conducted in an N₂ atmosphere to ensure device performance is maintained.

Through this optimized process, we successfully fabricated a 16 × 16 device array with high uniformity, confirming the practicality and effectiveness of our transfer and bonding optimization strategies. As shown in **Figure R11**, the array exhibits stable and uniform light-emitting behavior with a spatial resolution of 336 PPI. Moreover, the statistical analysis of the device parameters revealed excellent reproducibility, with the coefficient of variation for the on/off ratio and threshold voltage (V_{th}) remaining below 10%, underscoring the scalability and reliability of the optimized process. These results strongly support the potential of our integration scheme for medium- to large-scale array production, providing both theoretical and practical foundations for future intelligent display systems.

Figure R11 Uniformity characterization of HfO₂-based IPPMLED arrays. **a.** An overview optical micrograph and a high-magnification image of the fabricated HfO₂-based IPPMLED array. **b.** Uniform light emission from the operational 16×16 array, demonstrating four distinct luminance levels. **c.** Transfer characteristic curves of 100 individual HfO₂-based IPPMLED devices at a data voltage of 3 V. **d.** Statistical distributions of the V_{th} (i) and on/off state (ii) extracted from the transfer curves.

(2) Current status of 2D-TFT-driven micro-LED displays

We would also like to emphasize that research on 2D-material-based TFT backplanes driving micro-LED displays is still in its early stages. As we surveyed, those reported studies to date remain at the level of small-scale prototypes without achieving truly high-resolution or high-uniformity integration (see **Table R2**). This clearly indicates that numerous challenges related to **fabrication processes, large-scale integration, and material reliability** remain to be addressed in this emerging field.

On the other hand, we would like to emphasize that, different from previously reported works, the primary contribution of our work lies in proposing and experimentally validating a novel memory-compute-display integrated architecture. Thus, within this paradigm, the 5×5 and newly supplemented 16×16 arrays used in this study are sufficient to clearly prove the proposed pixel-level functions, such as data storage, computation, and in-situ display at the hardware level. This lays a foundational principle for future systems targeting high-resolution applications. We are confident that the proposed architecture can be extended to higher-resolution displays with continued optimization of transfer and integration processes.

Table R2 The surveyed studies on the array scale of vdW transistor-driven micro-LED arrays.

References	Display type	Driver transistors	Array scale	V_{data}	Maximum gate operating voltage
Nat. Nanotechnol. 17, 500–506 (2022).	Micro-LED	MoS ₂ transistor	16×16	8 V	10 V
Adv. Mater. 36, 2309531 (2024)	Perovskite Light-Emitting Diode (PeLED)	MoS ₂ transistor	8×8	6 V	15 V
Sci. Adv. 6, eabb5898 (2020).	OLED	MoS ₂ transistor	18×18	4V	9V
This work	Micro-LED	MoS₂ transistor	16×16	3 V	4V

Corresponding changes made in the manuscript:

In response to the reviewer’s comments, we have conducted a comprehensive review of the current research status of two-dimensional TFT-driven micro-LED displays and provided a systematic exploration of pathways for improving device uniformity in **Figure S13** and **Figure S14**. Furthermore, we have also supplemented the experimental data with the fabrication and characterization of 16×16 device arrays, and incorporated the statistical analysis of device uniformity into **Figure S15**.

“On the other hand, to meet the low-power requirements of edge computing applications, we further investigated the potential of high- κ dielectrics in replacing SiO₂ gate insulator for power consumption optimization (38, 39). **Figure 5a** shows the successfully fabricated 16×16 IPPMLED arrays with an increased pixel density of 336 PPI by reducing the pixel pitch. The detailed process optimization for large-scale array integration is provided in **Figure S13** and **Figure S14**, showcasing refined methods for large-area vdW material transfer and micro-LED bonding techniques. As expected, the IPPMLED device with dielectric engineering exhibits a notable reduction in the driving voltage, while maintaining good device-to-device uniformity in both V_{th} and on/off-state across 100 devices ($C_v=9.9\%$) (**Figure S15**).” on Page 10.

Comment #2: The current array performs contrast enhancement task, but practical image processing pipelines also require low-level (e.g., color correction, gamma correction), mid-level (e.g., segmentation, feature extraction), and high-level (e.g., recognition) processing. Please clarify how these additional functions would be handled by the in-pixel architecture versus off-chip.

Response:

We thank Reviewer #2 for raising this professional comment regarding the distribution of image processing tasks across our architecture. We totally agree with Reviewer #2’s suggestion that a complete image processing pipeline indeed involves multiple levels, from low-level to high-level operations, and the rational allocation of

these tasks among different hardware units is crucial for building an efficient image processing and display system.

In our work, the proposed IPPMLED architecture primarily leverages the intrinsic optoelectronic response and nonvolatile memory characteristics of the device to perform contrast enhancement and neural network inference tasks. To maximize overall energy efficiency and functionality, our design philosophy follows a hybrid partitioning approach—executing massively parallel, high-energy operations within pixels, while delegating precision or control-intensive computations to off-chip processors. **Specifically, contrast stretching and in-pixel matrix–vector multiplication are implemented in situ, whereas high-level functions such as deblurring or feature extraction are handled by an MCU (STM32F103RCT6) and external computing units. Below, we elaborate on this architecture and its hierarchical task allocation.**

(1) In-pixel implementation of low-level image processing

The contrast enhancement function demonstrated in this work is achieved entirely by exploiting the intrinsic piecewise-linear electroluminescent characteristics of the micro-LED pixels—without requiring any external computation unit. In practice, when an input voltage is directly applied to each pixel, the output luminance is governed by its input–output transfer curve, which includes distinct dead, linear, and saturation regions. This physical response naturally performs a nonlinear remapping of the input gray levels, aligning closely with the algorithmic principle of contrast stretching (as shown in **Figure R12**). Similarly, basic preprocessing functions such as gamma correction can be realized by tailoring the mapping curve between the input voltage and the pixel’s optical output. By integrating simple on-chip pre-processing circuits, the input signal can be conditioned and subsequently displayed through the IPPMLED array without external computation.

Figure R12 In-pixel implementation of low-level image processing. a. Schematic of the in-situ contrast stretching process. **b.** Flowchart of the IPPMLED-based display processing system, which integrates both the low-level and high-level processing modules.

(2) In-pixel/off-chip collaborative processing for high-level tasks

It is important to emphasize that higher-level tasks, such as decision-making and display calibration, are accomplished through cooperation between the IPPMLED array, the MCU, and an external computer. Specifically, our array features weight deployment and computing functions that are essential for executing energy-intensive matrix–vector multiplications during neural network inference.

As shown in **Figure R13**, we have established an IPPMLED-based hardware inference system for high-level image recognition, in which the array stores the trained network weights and executes the essential multiply–accumulate (MAC) operations. As illustrated in **Figure R13b-c**, trained and quantized neural network weights are programmed into the IPPMLED device conductance states. When an enhanced and encoded input voltage vector is applied to the array, Kirchhoff’s current law enables parallel analog computation of the matrix–vector multiplication within the array. This operation constitutes the most computation- and energy-intensive portion of neural network inference. Simultaneously, the same voltage stimuli directly drive the micro-LEDs, enabling in-situ display of the processed image. Subsequent tasks such as input timing control, analog-to-digital conversion (ADC), nonlinear activation, and final classification—are handled by the off-chip MCU and computer. **Figure R13d** presents an inference result with a recognition accuracy of 99.29%, demonstrating that this collaborative framework efficiently balances in-pixel parallelism with system-level control and precision, thus ensuring scalability and versatility across different image processing applications.

In summary, our IPPMLED architecture performs low-level image processing (e.g., contrast stretching) fully in situ within the array, and core MAC operations for high-level neural network recognition collaboratively with off-chip processors. The division of labor between in-pixel and off-chip processing not only improves energy efficiency but also enhances functional flexibility. Most importantly, the key innovation of our work lies in the architectural realization of an in-pixel processing display paradigm. While previous studies have largely focused on the memory or light-emission characteristics of individual devices (*Sci. Adv.* **8**, eabq4824 (2022)), our work, for the first time, demonstrates the **cooperative operation of storage, computation, and display at the pixel level within a hardware array**. This constitutes a fundamental step toward building energy-efficient intelligent display systems.

Figure 13 In-pixel/off-chip collaborative processing for high-level tasks. **a.** Photograph of the display processing platform, showing the matrix-configured IPPMLED array mounted on a custom PCB for multi-channel measurement. **b.** The weight distributions of an IPPMLED array from both post-training simulation and post-deployment. **c.** Current recordings from the IPPMLED array while executing MAC operations. **d.** Confusion matrices for the recognition of IPPMLED-enhanced images (i) versus original images (ii). **e.** Flowchart of the weight mapping and inference process, incorporating the IPPMLED array, peripheral electronics.

Corresponding changes made in the manuscript:

In response to the reviewer’s concerns, we have updated Figure 5 to provide a clearer illustration of the functional partitioning within the IPPMLED-based display system when handling various image processing tasks. We sincerely thank the reviewer once again for this insightful question, which has helped us clarify our design philosophy and the system-level task allocation in our architecture.

“In this work, the IPPMLED features gate-tunable luminance response and in-memory computing capability, offering significant advantages for in-situ display processing. To fully harness the advantages of the proposed devices, both contrast stretching and image

recognition are taken as classic examples to exhibit how the IPPMLED array provides a processing platform towards an optimal balance between in-pixel processing and off-chip management.” on Page 11.

“To comprehensively assess the capability of the proposed IPPMLED in reshaping the intelligent in-situ display landscape, a synchronized display processing pipeline capable of handling representative multi-level tasks has been designed. As shown in **Figure 5b**, the input image pixels are first converted into driving voltages according to the device-specific voltage–luminance characteristic, which intrinsically aligns with the contrast-stretching algorithm. In this scheme, low pixel values are mapped to the sub-threshold voltage region (no emission), high values to the saturation region, and intermediate values to the linearly stretched operating region, respectively, thereby enabling immediate contrast-enhanced display. To further optimize the display output, an in-pixel recognition function is proposed. Specifically, the IPPMLED array, serving as the neural network accelerator, enables efficient hardware inference by executing batched weight deployment and in-situ MAC operations. The system subsequently determines whether to trigger MCU-assisted image deblurring according to the obtained recognition results. For instance, if the obtained recognition accuracy falls below a predefined threshold, an advanced deblurring procedure would be activated with the assistance of the MCU (STM32F103RCT6) and back-end processors. Conversely, a higher recognition accuracy would bypass high-level processing to achieve the high-quality display.” on Page 12.

“The quantitative assessment is further considered to validate the contrast stretching operation, providing better guidance for subsequent refinement of the display architecture. We demonstrate an in-pixel image recognition using an IPPMLED-based feedforward neural network, applied to both original noisy letter images and their IPPMLED-enhanced counterparts (Figure 5e). As shown in Figure 5f, the trained weights are extracted and deployed onto the IPPMLED array via non-volatile conductive retention, achieving a complete correspondence with the target weight matrix. Subsequently, the letter images (5×5 pixels) are transformed into 25×1 vectors to serve as inputs, where parallel MAC operations are performed with the deployed weights, enabling highly parallel, low-power analog computing and in-situ execution of the inference process within the IPPMLED. Figure 5g illustrates a representative current summation process across the array during this operation. The resulting MAC values are temporarily stored in the readout circuit and buffer registers, after which weight refreshing and array reuse enable the processing of subsequent batches, ultimately yielding the recognition result.” on Page 12.

Comment #3: The main limitation of trap-based synaptic devices is slow processing, which may cause image ghosting. In Fig. 4a-c, the postsynaptic current appears to stabilize over tens of seconds, which conflicts with high refresh rate operation. Please report the relevant time constants at the operating point and demonstrate video-rate updates without image persistence. If the display mode does not rely on slow synaptic effects, please state this explicitly and provide measurements that confirm millisecond scale settling suitable for typical display rates of 60 to 240 Hz.

Response:

We sincerely appreciate the reviewer's insightful comments regarding the operational speed of trap-based synaptic devices and its potential impact on display performance. The charge-trapping mechanism indeed serves as the fundamental operating principle of our IPPMLED. However, **it is important to emphasize that the synaptic plasticity of the device can be modulated by varying the gate voltage conditions—specifically, long-term plasticity (LTP) for non-volatile memory and short-term plasticity (STP) for rapid switching.** The long-term plasticity observed under high-voltage, wide-pulse conditions is intentionally utilized for non-volatile weight mapping in neural network deployment and inference. In contrast, during display refresh operations, the device operates under low-voltage, short-pulse conditions, where it exhibits short-term plasticity that readily supports high-frequency switching, thereby fully accommodating standard display refresh rates. To clarify this distinction, we have systematically revised the manuscript and supplemented it with relevant experimental data and video demonstrations.

Table R3 Survey on synaptic plasticity behaviors and application scenarios of emerging synaptic devices.

References	Device	STP stimulation	LTP stimulation	Applications
Adv. Mater. 37(17), 2412006.	Au/Cr/BTO/NSTO memristor	Single pulse (pulse width: 1 μ s)	Single pulse (pulse width: 1 ms)	Neural network calculations
Nat. Commun. 16, 9506 (2025)	MoS ₂ memtransistor	Single pulse (pulse width :100 ms)	Single pulse (pulse width: 1 s)	Neural network calculations
Nat. Commun. 12, 2480 (2021)	Organic electrochemical transistor (OECT)	Single pulse (pulse width: 20 ms)	Multiple pulses, (pulse width: 10 ms)	Associative learning
Adv. Sci. 11, 2305679 (2024)	SnS ₂ /h-BN/CIPS-based Fe-FET	Multiple pulses (pulse number: 10)	Multiple pulses (pulse number:50)	Associative learning
Sci. Adv. 8, eabq4824 (2022)	Synaptic organic light-emitting transistors (OLET)	Single pulse (pulse width: 6 s)	Single pulse (pulse width: 12 s)	Memory display
This work	MoS ₂ transistor	Single pulse (pulse width: 1 ms)	Single pulse (pulse width: 500 ms)	Display processing

(1) Long-term and short-term plasticity in synaptic response

As widely reported in studies on synaptic devices, the plasticity behavior of charge-trap-based synaptic transistors strongly depends on the electrical pulse conditions (*Nat. Commun.* 15, 7671 (2024), *Nat. Commun.* 11, 2972 (2020)). To systematically illustrate this key characteristic, we have summarized the typical synaptic response patterns under different pulse conditions in **Table R3**, which clearly shows a continuous transition from volatile short-term plasticity to non-volatile long-term plasticity, along

with their respective application scenarios. Under high gate voltage and wide pulse width, a large number of carriers are injected into deep-level traps within the dielectric layer. These deeply trapped charges remain stable after the pulse is removed, enabling non-volatile long-term plasticity. This behavior is particularly suitable for implementing non-volatile weight storage and updating in artificial neural networks (*Nat. Electron.* **5**, 859–869 (2022), *Nat. Commun.* **13**, 1670 (2022)). Conversely, under low gate voltage and short pulse width, only shallow-level traps participate in fast and reversible charge dynamics, resulting in volatile short-term plasticity. This rapid, transient dynamic is well-suited for applications requiring quick response and short-term memory, such as real-time signal processing.

(2) Clarification of synaptic effects and display refresh mechanism

Based on the distinct plasticity behaviors under different voltage conditions, our architecture explicitly separates the weight mapping mode from the display refresh mode at the system level, thereby enabling the co-existence of in-memory computing and high-speed display.

The non-volatile postsynaptic current lasting tens of seconds, as observed in Figures 4a–c, is intentionally designed and utilized for in-memory computing and neural network weight mapping. In this mode, pre-trained weights are programmed into the device conductance as non-volatile states through high-amplitude, long-duration electrical pulses (**Figure R14a-b**). This process solidifies the core network weights within the pixel array, enabling subsequent analog MAC operations to be performed in situ and in parallel. It is important to note that weight mapping is only performed during system initialization or task reconfiguration and does not require high-speed writing.

Figure R14 Long-term and short-term synaptic plasticity of IPPMLEDs. a,b. Synaptic plasticity under both negative (a) and positive gate-voltage pulses (b) with varying pulse widths, showing memory retention up to 150 seconds. **c,d.** Current response characteristics of the IPPMLED at pulse frequencies of 120 Hz (c) and 240 Hz (d), demonstrating the rapid refreshing potential of the proposed pixel driving design.

During display refresh, the luminance of the IPPMLED is modulated by applying fast, low-amplitude electrical pulses that induce short-term plasticity (**Figures R14c-d**). Under these conditions, the charge dynamics remain entirely within the fast, reversible regime, with a response speed significantly shorter than the frame period required for conventional displays. This ensures that the luminance of each pixel can be updated rapidly and without retention, fully meeting the refresh rate requirements of 60–240 Hz.

(3) Experimental validation of video-rate refresh operation

To directly demonstrate the video-rate refresh capability of our device in display mode and the absence of image persistence, we conducted the suggested supplementary experiments, focusing on the dynamic performance of the device at refresh rates of 60 Hz, 120 Hz, and 240 Hz. As shown in **Videos R1-6**, the IPPMLED exhibits a millisecond-scale rapid response to short input pulses. It should be noted that these videos are provided both in 1× and 10× slow-motion modes for better observation (please identify different videos via the defined names). **Specifically**, due to the inherent limitations of smartphone cameras, which often result in insufficient sampling points during recording, the screen refreshing phenomena are not clearly visible in videos captured at normal playback speed. To mitigate this issue, a 10× slow-motion mode is provided, which enhances temporal resolution and allows for better observation of rapid changes. This phenomenon is more pronounced between the 1× and 10× slow-motion modes at 60 Hz. Although the severe undersampling at 240 Hz is not fully compensated by the 10× slow-motion mode, flickering phenomena can still be observed.

In summary, by strategically designing the electrical operating scheme, our IPPMLED successfully decouples non-volatile weight storage for in-memory computing from high-speed luminance refresh for display. The supplementary experimental data further demonstrate its compatibility with high-performance video display at 60–240 Hz. We deeply appreciate the reviewer’s insightful comments, which have greatly helped us improve this work. We hope that the above explanations and newly provided data adequately address the concerns raised.

Corresponding changes made in the manuscript:

In response to the reviewer’s suggestions, we have supplemented the **main text** with a detailed explanation of the distinct operational mechanisms under weight mapping and display refresh modes, in order to avoid possible misunderstanding caused by the previous description. Additionally, we have conducted dynamic performance tests at refresh rates of 60 Hz, 120 Hz, and 240 Hz (**Figure S6** and **Figure S18**).

“In summary, the IPPMLED device exhibits diverse conductance plasticity characteristics under different pulse conditions, enabling versatile display operating modes.” on Page 9.

“Similarly, ideal nonlinear or triplet-stage voltage-dependent luminescence characteristics along with non-volatile memory behavior can be reproduced in the IPPMLED device with HfO₂ dielectric engineering. As illustrated in **Figure S17a**, when scanned under gate voltages ranging from -4 V to $+4$ V, the device exhibits three well-defined emission regions characterized by a non-emissive dead zone below 0 V, an approximately linear response, and a saturation region at intermediate and higher voltages (above 3 V), respectively. Furthermore, the device exhibits controllable conductance modulation, akin to the flexible synaptic plasticity, under different pulse programming conditions (Figures S17b-e). Specifically, the conductance state demonstrates non-volatile conductance retention lasting over 150 seconds with wider-pulse stimulus, highlighting its potential for neural network weight deployment applications. In contrast, the device achieves instantaneous refresh responses under short pulse widths, such as at frequencies of 60 Hz, 120 Hz, and 240 Hz, emphasizing its capability for display refreshing (**Figure S18**).” on Page 10.

Comment #4: Please expand the micro-LED section to include fabrication flow and key device metrics such as luminance, current density, and EQE to clarify emitter performance.

Response:

We sincerely thank Reviewer #2 for pointing out the clarify the detailed fabrication and performance of the employed micro-LED in this work. As the core light-emitting component of the IPPMLED, we fully agree that a detailed description of its fabrication process and quantitative data on luminance, current density, and external quantum efficiency (EQE) are essential for clarifying the overall performance of the IPPMLED. Following the reviewer’s advice, we have supplemented the following information:

(1) Fabrication process of the micro-LED array

This work employs a discrete flip-chip micro-LED process, which involves the following key steps (**Figure R15a**):

- (i) Metal-organic chemical vapor deposition (MOCVD) was used to grow an n-type GaN layer (Si-doped), an InGaN/GaN multiple quantum well (MQW) light-emitting layer, and a p-type GaN layer (Mg-doped) sequentially on a sapphire substrate. Inductively coupled plasma (ICP) dry etching was then applied to remove the p-GaN and MQW layers, exposing the n-GaN layer.
- (ii) An indium tin oxide (ITO) current-spreading layer was deposited on the GaN surface via sputtering, followed by ICP etching to pattern the ITO. An Ag reflective layer was subsequently formed by electron-beam evaporation to enhance top-side light extraction.
- (iii) A SiO₂ isolation layer was deposited over the micro-LED structure by plasma-enhanced chemical vapor deposition (PECVD). Dry etching was performed to open

vias in the SiO₂, exposing the underlying Ag reflector and n-GaN layer. Finally, patterned metal electrodes for individual pixels were formed by electron-beam evaporation on both the mesa and n-GaN regions.

Figure R15b-c presents a schematic of the fabricated discrete flip-chip micro-LED array and a scanning electron microscopy (SEM) image of a single pixel, clearly showing the pixel dimensions and pitch.

Figure R15 Fabrication process and optical imaging of micro-LEDs. a. Schematic illustration of the micro-LED fabrication process. **b.** Optical micrograph of a single micro-LED device. **c.** Optical micrograph of a micro-LED array, illustrating the device layout and pitch.

(2) Key performance metrics of the micro-LED

To clarify the performance of the micro-LED emitting unit, we have added key electro-optical characterization (**Figure R16**) in the revised manuscript, including luminance, current density, and EQE.

In terms of current density, the device reaches a luminance of 3.39 cd/m² at a current density of 312.5 A/cm², demonstrating excellent high-current injection capability and high luminance potential. Regarding external quantum efficiency, the device reaches a peak EQE of 32.46% at a current of 0.29 mA. These experimental metrics confirm that the fabricated micro-LED is a high-performance light-emitting device capable of meeting the demanding requirements of the proposed display-processing integration application.

Figure R16 Electroluminescence performance of the micro-LED. a. Relationship between the luminance and injection current of the micro-LED. **b.** External quantum efficiency characteristics of the micro-LED. **c.** Relationship between the current and luminance of the micro-LED.

Corresponding changes made in the manuscript:

Based on Reviewer #2’s suggestions, the fabrication flow and key metrics for the micro-LED devices have been provided in the Supporting Information (**Note S1, Figure S2, and Figure S3**). We sincerely thank the reviewer again for prompting this clarification, which significantly enhances the completeness of the paper.

“The blue-emitting micro-LED devices in this architecture are fabricated on the mature gallium nitride (GaN) platform (**Figure S2 and Note S1**), which offers significant advantages in material stability, electro-optical efficiency, and operational lifetime (36, 37). The micro-LED cell exhibits remarkable luminescence performance, achieving a peak external quantum efficiency (EQE) of 32.46% at a current of 0.29 mA, and a luminance of 3.13×10^6 cd m⁻² at a current density of 2.4×10^5 mA cm⁻² (**Figure S3**).” on Page 5.

“The GaN-based blue LEDs are fabricated beginning with the metal-organic chemical vapor deposition (MOCVD) growth of n-GaN, InGaN/GaN multiple quantum well (MQW), and p-GaN epitaxial layers on sapphire substrates. The mesa structure is then defined by inductively coupled plasma (ICP) etching to expose the n-GaN layer, followed by sputtering and patterning of an indium tin oxide (ITO) current spreading layer. An Ag reflective layer is subsequently deposited via electron-beam evaporation to enhance light extraction efficiency. Device isolation is achieved through plasma-enhanced chemical vapor deposition (PECVD) of a SiO₂ layer with via openings formed by dry etching. Finally, patterned metal electrodes are fabricated by electron-beam evaporation to complete the pixel array. A detailed flowchart of the fabrication process is provided in **Figure S2**.” in the Method.

Comment #5: In Figure 2e, the variation of threshold voltage appears large for display-class uniformity, which could limit high-resolution operation. Please describe compensation methods and pathways to enhance the uniformity.

Response:

We sincerely appreciate the reviewer’s valuable comments regarding threshold voltage uniformity, which is indeed critical for high-resolution display applications. We

acknowledge that the V_{th} variation of 11.98% shown in Figure 2e, while reasonable for early-stage development, requires further improvement to meet the stringent demands of commercial displays. In response, we have implemented optimizations in both material and device fabrication, and validated their effectiveness through statistical analysis of larger device arrays.

(1) Optimization strategies for improved uniformity

In vdW transistor-based devices, the uniformity is strongly influenced by intrinsic material quality and process-induced variations, such as material crystallinity, transfer integrity, and interface properties. Therefore, recent research has focused on co-optimizing material growth, damage-free transfer, and key integration steps to improve device-to-device consistency.

For material optimization, the key lies in enhancing large-area uniformity and electrical stability in MoS₂ films. Advanced approaches in the field include precise control of chemical vapor deposition parameters—specifically growth temperature and precursor gas flow—to optimize film crystallinity, continuity, and homogeneity, complemented by post-growth annealing for effective defect reduction (*Nature* **642**, 327–335 (2025)). Additionally, damage-free transfer of two-dimensional materials is a widely recognized research priority, given its essential role in enhancing the uniformity of integrated device arrays. For example, Prof. Jing Kong’s team demonstrated an advanced transfer technique based on electrostatic repulsion, using an ammonia solution compatible with CMOS industry standards for etch-free, high-yield, wafer-level MoS₂ transfer (*Nature* **645**, 906–914 (2025)).

In terms of device unit integration and fabrication optimization, the key challenge is achieving high-quality, reliable bonding between micro-LEDs and driver transistors. Specifically, the selected bonding technique along with its critical process parameters (including bonding temperature, pressure, interfacial flatness, and alignment accuracy), directly affects multiple core device performance metrics (*Research*, **6**, 47 (2023); *Light Sci. Appl.* **9**, 83 (2020)). Notably, mechanical stress and thermal budget during bonding significantly influence the electrical characteristics of vdW transistors, thereby impacting overall device uniformity and stability. In summary, high-quality large-area 2D material growth, damage-free transfer, and precision bonding between micro-LEDs and transistors have been established as effective pathways to suppress performance variations caused by process deviations.

(2) Experimental verification of uniformity improvement

To objectively assess the effectiveness of the optimization pathways mentioned above within our laboratory setup, we adopted large-area wet transfer of two-dimensional materials and low-temperature thermocompression bonding for micro-LEDs in the newly supplemented experiments. Specifically, for the large-area wet transfer of two-dimensional materials, we ensured material integrity and reduced surface damage by treating hydrophilic substrates and gently removing PMMA using benzyl ether. For optimizing the micro-LED bonding process, we used low-melting-

point bonding metals to prevent high-temperature loss of materials, while maintaining an inert nitrogen (N_2) atmosphere to further protect the materials.

As a result, by implementing these process optimization strategies, we successfully fabricated large-scale device arrays (16×16 arrays) that exhibited uniform and stable light emission under different operating voltages (**Figure R17 a-b**). Furthermore, we conducted a comprehensive statistical analysis of their threshold voltages (**Figure R17 c-d**). The test results demonstrated that the coefficient of variation for the device threshold voltage was reduced to below 10%, confirming the effectiveness of the proposed optimization approach.

We sincerely thank the reviewer for this valuable feedback, which has helped us not only to outline a clear pathway for uniformity enhancement but also to experimentally validate its feasibility.

Figure R17 Uniformity characterization of HfO_2 -based IPPMLED arrays. **a.** Optical micrograph of the IPPMLED array based on the HfO_2 dielectric layer, noting that the image appearance depends on specific imaging conditions and microscope settings. **b.** Uniform light emission from the operational 16×16 array, demonstrating four distinct luminance levels. **c.** Transfer characteristic curves of 100 individual HfO_2 -based IPPMLED devices at a fixed data voltage of 3 V. **d.** Statistical distributions of the V_{th} (i) and on/off state (ii) extracted from the transfer curves.

Corresponding changes made in the manuscript:

To address the reviewer's feedback, we have conducted a comprehensive review of strategies for enhancing device uniformity, as detailed in **Figures S13-S14**. Furthermore, we have supplemented the experimental data by fabricating and

characterizing a 16×16 device array. The statistical analysis of its uniformity is now included in **Figure S15**.

“On the other hand, to meet the low-power requirements of edge computing applications, we further investigated the potential of high- κ dielectrics in replacing SiO₂ gate insulator for power consumption optimization (38, 39). **Figure 5a** shows the successfully fabricated 16×16 IPPMLED arrays with an increased pixel density of 336 PPI by reducing the pixel pitch. The detailed process optimization for large-scale array integration is provided in **Figure S13** and **Figure S14**, showcasing refined methods for large-area vdW material transfer and micro-LED bonding techniques. As expected, the IPPMLED device with dielectric engineering exhibits a notable reduction in the driving voltage, while maintaining good device-to-device uniformity in both V_{th} and on/off-state across 100 devices ($C_v=9.9\%$) (**Figure S15**). Correspondingly, the inherently low power consumption of the synaptic transistor translates into a great energy efficiency advantage for the integrated LED device over current display technologies (**Figure S16 and Table S2**).” on Page 6.

Comment #6: Only blue micro-LED devices are demonstrated. Please discuss how the in-pixel processing systems scale to multi-color RGB displays.

Response

We sincerely appreciate the reviewer’s insightful observations regarding the scalability of our in-pixel processing system to multi-color RGB displays, which provides a valuable perspective for the completeness and future direction of our work. Below, we elaborate on the foundational significance of the current blue monochromatic demonstration, feasible technical pathways for extending to full-color RGB displays, and the associated challenges.

(1) Significance of the blue monochromatic demonstration

This proof-of-concept study employed blue micro-LEDs, leveraging their well-established GaN material system for its superior stability and electro-optical efficiency (*Nat. Electron.* **5**, 859–869 (2022); *Nat. Commun.* **14**, 1386 (2023)). This choice allowed the research to focus on validating the core operational principle of the memory-compute-display architecture, without the additional complexities associated with less mature material systems. Furthermore, the core mechanism of our architecture lies in the integration strategy between the synaptic transistor and the micro-LED, which is fundamentally agnostic to the specific bandgap (i.e., emission color) of the active light-emitting layer. As such, the architecture possesses an inherent physical basis and potential for extension to other colors.

(2) Feasible technical pathways for RGB integration

While this work focuses on the first demonstration of a vdW transistor-driven micro-LED pixel unit and the validation of its display-memory-compute fusion functionality, we greatly appreciate the reviewer’s suggestion regarding full-color displays, which provides innovative direction for our subsequent research. We have

analyzed feasible technical routes for achieving this goal. Specifically, full-color RGB displays could be realized through massive transfer integration, where red, green, and blue micro-LED chips are accurately assembled onto a common backplane. This technology is already widely used in commercial micro-LED displays, indicating high process feasibility (*Light Sci. Appl.* **9**, 83 (2020); *Light Sci. Appl.* **12**, 258 (2023)). The core innovation of our study lies in the integration of a signal-processing synaptic transistor within each pixel. This processing unit is color-agnostic and can drive and control RGB sub-pixels in parallel. Therefore, image processing functions demonstrated in the current monochromatic system, such as contrast stretching, can in principle be directly extended to full-color systems. By independently modulating the three sub-pixels, hardware-level processing of color images can be achieved. To realize RGB integration, independent driving and modulation pathways for the three sub-pixels would need to be designed at the circuit level, and a color management module would need to be incorporated into the image mapping algorithm.

(3) Key challenges in realizing full-color RGB displays

Although full-color RGB display is theoretically feasible, several engineering challenges remain. First, micro-LED devices of different colors (e.g., GaN-based blue/green emitters and InGaN/AlGaInP-based red emitters) exhibit significant differences in material systems, external quantum efficiency, and wavelength stability. Compensating for these variations on a common backplane through driving strategies to ensure brightness and chromaticity uniformity across the entire RGB display constitutes a complex system-level challenge. Second, integrating three sub-pixels and their associated driving circuits within a limited pixel area imposes constraints on pixel density improvement, necessitating further breakthroughs in device miniaturization and integration processes. Additionally, preventing optical and electrical crosstalk between different color sub-pixels requires careful device layout design.

Overall, we acknowledge that extending the current system to full-color RGB display is a critical yet challenging objective for the next research phase. On the other hand, the current monochromatic system based on blue micro-LEDs in this work has successfully established a functional and principled foundation for such an extension. Therefore, we once again thank the reviewer for their insightful comments, which have provided clear guidance for our future research direction.

Corresponding changes made in the manuscript:

In response to the reviewer's valuable comments, we have supplemented the **main text** with an explanation of the rationale behind selecting blue LEDs for this experimental study.

“The blue-emitting micro-LED devices in this architecture are fabricated on the mature gallium nitride (GaN) platform (**Figure S2 and Note S1**), which offers significant advantages in material stability, electro-optical efficiency, and operational lifetime (36, 37). The micro-LED cell exhibits remarkable luminescence performance, achieving a peak external quantum efficiency (EQE) of 32.46% at a current of 0.29 mA, and a

luminance of $3.13 \times 10^6 \text{ cd m}^{-2}$ at a current density of $2.4 \times 10^5 \text{ mA cm}^{-2}$ (Figure S3).” on Page 5.

Comment #7: In Fig. 3b, it seems that the required gate voltages are quite high. Is this common, and realistic for actual display operation? If not, are there ways/roadmaps to reduce this?

Response:

The authors sincerely appreciate the reviewer’s insightful comments regarding the operating gate voltages observed in our study. We acknowledge that the relatively high gate voltages required for devices with SiO₂ dielectric could present challenges for practical display applications, and we fully agree that reducing the operating voltage is essential for improving energy efficiency and practical implementation. Accordingly, we have systematically investigated and implemented a dielectric engineering approach to address this challenge, with key results presented below.

(1) Rationale for dielectric material selection

In the initial proof-of-concept phase, a 300 nm SiO₂ layer was selected as the gate dielectric primarily owing to its ability to provide excellent color contrast for transferred vdW materials. This choice allowed us to focus on validating the fundamental operational principles of the novel memory-compute-display architecture while ensuring experimental reproducibility and stability, although it necessarily entailed compromises in operating voltage and power consumption. After successfully demonstrating the basic functionality of the IPPMLED unit, we proceeded to develop and integrate a high-κ HfO₂ dielectric process specifically to address the low-power requirements essential for edge computing applications.

16 IPPMLED array based on the HfO₂ dielectric layer. **b.** The corresponding uniform light emission image from the operational 16×16 array. **c.** Transfer characteristic curves of 100 individual HfO₂-based IPPMLED devices. **d.** Statistical distribution of V_{th} from transfer characteristic curves at a fixed data voltage of 3 V. **e.** A comparison table benchmarking the operating voltages of this work against current representative micro-LED technologies.

(2) Roadmaps for reducing operating gate voltage

Following the reviewer's suggestion, we explored practical approaches to reduce the operating voltage. HfO₂, with its substantially higher dielectric constant (~20–25) compared to SiO₂ (~3.9), enables a significantly larger gate capacitance per unit area at the same physical thickness, thereby effectively improving the subthreshold swing and reducing the transistor's turn-on voltage (*Chem. Rev.* **118**, 5690-5754 (2018); *Nat. Commun.* **11**, 6207 (2020)). Drawing on well-established approaches from advanced node integrated circuits, we deposited high-quality HfO₂ films via atomic layer deposition (ALD) as the gate dielectric. This enabled the successful fabrication of a 16×16 IPPMLED array based on HfO₂ (**Figure R18a-b**). Statistical analysis of the driving voltages across the array demonstrates that this optimization successfully reduced the operating voltage from –35 V to –0.22 V, to levels below those of existing display technologies (**Figure R18c-e**). These results confirm the effectiveness and feasibility of the high- κ dielectric approach for achieving low-voltage, high-uniformity large-array operation.

We sincerely thank the reviewer for this valuable comment. Your observation has greatly helped us to enhance the stringency and completeness of this study by prompting us to develop and validate a clear roadmap toward low-voltage operation.

Corresponding changes made in the manuscript:

In response to the reviewer's valuable comments regarding the operating voltage, we have supplemented the main text with a study on experimental methods for reducing the driving voltage, as detailed in **Figure S13** and **Figure S14**. Furthermore, we have fabricated a large-scale device array (**Figure 5a**) and provided the corresponding statistical analysis of the operating voltage in **Figure S15**.

“Note that 300 nm SiO₂ is employed as the gate dielectric during the early-stage exploration and validation of the proposed display architecture, owing to its advantages in better color contrast for transferred vdW materials and hence improving the processing reliability.” on Page 4.

“On the other hand, to meet the low-power requirements of edge computing applications, we further investigated the potential of high- κ dielectrics in replacing SiO₂ gate insulator for power consumption optimization (38, 39). **Figure 5a** shows the successfully fabricated 16×16 IPPMLED arrays with an increased pixel density of 336 PPI by reducing the pixel pitch. The detailed process optimization for large-scale array integration is provided in **Figure S13** and **Figure S14**, showcasing refined methods for large-area vdW material transfer and micro-LED bonding techniques. As expected, the IPPMLED device with dielectric engineering exhibits a notable reduction in the driving

voltage, while maintaining good device-to-device uniformity in both V_{th} and on/off-state across 100 devices ($C_v=9.9\%$) (**Figure S15**). Correspondingly, the inherently low power consumption of the synaptic transistor translates into a great energy efficiency advantage for the integrated LED device over current display technologies (**Figure S16 and Table S2**).” on Page 10.

“Similarly, ideal nonlinear or triplet-stage voltage-dependent luminescence characteristics along with non-volatile memory behavior can be reproduced in the IPPMLED device with HfO₂ dielectric engineering. As illustrated in **Figure S17a**, when scanned under gate voltages ranging from -4 V to $+4$ V, the device exhibits three well-defined emission regions characterized by a non-emissive dead zone below 0 V, an approximately linear response, and a saturation region at intermediate and higher voltages (above 3 V), respectively. Furthermore, the device exhibits controllable conductance modulation, akin to the flexible synaptic plasticity, under different pulse programming conditions (**Figures S17b-e**).” on Page 10.

Comment #8: The demonstrated pixel density of ~ 300 PPI is insufficient for microdisplays which typically require much higher PPIs (>2000) to reduce screen door effects. Therefore, the references to microdisplays in the abstract and introduction seem a bit misleading.

Response:

We thank the reviewer for raising this important point regarding pixel density. Based on industry research, current market demands specify pixel density requirements of approximately 90–280 PPI for television displays, 380–500 PPI for smartphones, and over 2000 PPI for near-eye displays such as AR/VR devices (*Light Sci. Appl.* **7**, 17168 (2018); *Light Sci. Appl.* **9**, 105 (2020)). We acknowledge that while the demonstrated resolution of ~ 300 PPI in this work meets the requirements for most conventional display applications, it remains below the typical specifications for high-resolution microdisplays targeted at AR/VR systems. The identified gap primarily stems from two key technical challenges in high-density integration: maintaining the electrical uniformity of two-dimensional materials during large-area transfer, and overcoming the limitations of high-precision bonding at micrometer-scale pixel pitches.

We have systematically explored viable pathways for enhancing integration density and conducted supplementary experiments to effectively increase the PPI. Under our laboratory conditions, we implemented optimizations in both material transfer and integration processes. For 2D material transfer, we adopted an improved large-area wet transfer method featuring hydrophilic substrate treatment and a gentle PMMA removal process using anisole, which effectively preserved material integrity and minimized surface damage. In parallel, we optimized the micro-LED pitch, reducing it from 84.6 μm to 43 μm . This was combined with a bonding strategy employing low-melting-point metals and an inert nitrogen atmosphere, ensuring improved integration density while maintaining satisfactory electrical uniformity. Employing these approaches, we have successfully fabricated a 16×16 array. This achievement has allowed us to raise the PPI from 300 to 336. (**Figure R19**).

Figure R19 IPPMLED array with a focus on PPI enhancement. **a.** An overview optical micrograph of the fabricated HfO₂-based IPPMLED array. **b.** A higher-magnification image corresponding to the region marked in (a), revealing the detailed device layout and spacing. **c.** An overview optical micrograph of the fabricated SiO₂-based IPPMLED array. **d.** A higher-magnification image corresponding to the region marked in (c), revealing the detailed device layout and spacing.

In summary, we have improved the pixel density through optimized material transfer and bonding processes, achieving a resolution that meets the requirements of mainstream computer and mobile displays. Furthermore, in accordance with the Reviewer #2's suggestion, we have carefully revised the abstract and introduction in the manuscript to remove any potentially misleading references to 'microdisplays', thereby more accurately reflecting the scope and current stage of our work (as below).

We would like to emphasize that the pixel size in this study was chosen primarily for proof-of-concept purposes. It provides a feasible platform for the first-time demonstration of a van der Waals transistor-driven micro-LED architecture with in-pixel memory and computing capabilities, balancing fabrication complexity with testing and characterization requirements. Future work will focus on overcoming the density bottleneck through advanced lithography, wafer-level heterogeneous integration, and other process innovations.

We are truly grateful for the reviewer's insightful comments, which have helped us better articulate the positioning and value of our current work while providing clear direction for our subsequent research.

Corresponding changes made in the manuscript:

Based on the reviewer's suggestions, we have revised the wording related to micro-displays in both the **abstract** and **introduction**. We are truly grateful for the reviewer's insightful comments, which have helped us better articulate the positioning and value of our current work while providing clear direction for our subsequent research.

“The rapid evolution of intelligent display technologies is driving the development of next-generation edge smart systems, yet conventional off-pixel processing architectures suffer from severe display latency and energy bottlenecks.” in the Abstract.

“The rapid advancement of intelligent display technologies is revolutionizing interactive experiences and enabling next-generation Internet of Things (IoT) applications (1–3).” in the Introduction.

“The integrated display approach in this work delivers a closed-loop solution that unifies image optimization and display within a compact, highly efficient pixel cell, advancing in-pixel image processing for next-generation intelligent display technologies.” in the Introduction.

“This capability critically addresses the need for enhanced visual smoothness in modern intelligent display technologies.” on Page 8.

Comment #9: In Table S1, I think it could be better to also display the integration density in PPIs since it's more commonly used for displays than units/cm².

Response:

We thank the reviewer for this valuable suggestion. We fully agree that PPI serves as a more intuitive and universally adopted metric than integration density (units/cm²) for evaluating display technologies. Following the reviewer's advice, we have now supplemented **Table S1** in the revised manuscript with corresponding PPI values alongside the original integration density data.

This addition enables more direct cross-comparison and assessment of performance parameters in line with standard practices in the display research community. We believe this revision significantly improves the clarity and professionalism of our data presentation.

Comment #10: The authors should provide a more thorough analysis of the device's power consumption, comparing that of IPPMLED to conventional off-pixel display system. This would provide a more complete picture of the energy efficiency benefits.

Response:

We sincerely appreciate the reviewer's insightful comments regarding power consumption analysis. We fully agree that a thorough comparison between our IPPMLED and conventional off-pixel display systems is essential for evaluating the true energy efficiency benefits. In response, we have conducted systematic power measurements and benchmarking to provide a more complete assessment.

(1) Intrinsic low-power characteristics of vdW synaptic transistors

The vdW synaptic transistor employed in this work, based on a two-dimensional MoS₂ channel, offers inherent advantages for low-power operation. Its dangling-bond-free vdW interface effectively suppresses carrier scattering, while the atomic-layer channel thickness enables superior electrostatic control, providing an ideal physical foundation for energy-efficient synaptic operation (*Nat. Nanotechnol.* **6**, 147–150 (2011)). To quantitatively evaluate the power consumption in neuromorphic computing applications, we calculated the energy per synaptic event for driver transistors using the formula $E = I_{\text{peak}} \times V \times \Delta t$. Measurements show that the energy consumption of our vdW driver transistor achieves 0.14 pJ. Benchmarking against recently reported synaptic transistors confirms that our HfO₂-based vdW devices achieve competitive energy efficiency (**Figure R20**), demonstrating a clear pathway for further optimization.

Figure R20 Energy consumption advantages of the IPPMLED. Benchmarking of power consumption per spike across various emerging synaptic electronic devices with different channel materials^[1-10].

(2) Overall power consumption of IPPMLED integrated units

To address application-level power concerns, we evaluated the operating power of the fully integrated IPPMLED unit. The device operates at $V_{\text{data}} > 3$ V, exhibiting a power consumption of 14.4 μW under a current density of 95.1 A/cm². **Table R4** provides a systematic analysis of recent advances in the LED field, including operating voltage and energy consumption, demonstrating the comprehensive advantages of our IPPMLED. Note that the limited test conditions presented in the surveyed studies preclude a unified statistical analysis of power consumption. Moreover, the vdW synaptic transistor-driven IPPMLED features programmable non-volatile conductance, providing another key energy-saving advantage for display processing. Once programmed, the conductance state is inherently retained, eliminating the need for power-intensive external components like constant current sources. This approach drastically cuts the power overhead from data movement and peripheral circuitry.

It is worth noting that the total system-level energy consumption of our IPPMLED array is challenging to estimate accurately, as it is significantly influenced by the input pulse sequences and the power overhead of peripheral circuits, such as analog-to-digital converters (ADCs) and microcontrollers (MCUs). This is because the array is utilized

at the hardware level for weight deployment and the implementation of MAC operations. This challenge in system-level energy assessment is also evident in previously published display driver studies (for example, *Nat. Nanotechnol.* **16**, 1231–1236 (2021), *Nat. Nanotechnol.* **17**, 500–506 (2022)), where a standardized framework for quantifying the total energy consumption of display systems is still lacking. Despite this, the low power consumption demonstrated at the device level lays a solid foundation for building future energy-efficient intelligent display systems.

Table R4 The surveyed studies about the emerging display architectures and comparative analysis of their power consumption.

References	Display type	Driver transistor	V_{data}	V_{gate}	Power consumption
Nat. Nanotechnol. 16, 1231-1236 (2021)	Micro-LED	MoS ₂ driver transistor	8 V	8 V	18 μ W (28 A/cm ²)
Nat. Electron. 6, 216-224 (2023).	Micro-LED	Poly-Si transistor	-5 V	-8 V	1345 μ W
Displays 87, 102997 (2025).	Micro-LED	Si-CMOS	4 V	4.5 V	30 μ W
Adv. Mater. 37, 2416015 (2025).	Micro-LED	Poly-Si transistor	-4 V	-10 V	~190 μ W (31.6 A/cm ²)
Adv. Mater. 37, 2411999 (2025).	Micro-LED	/	2.8 V	/	120 μ W
Nano Energy 135, 110613 (2025).	Micro-LED	/	6 V	/	361 μ W
Nat. Commun. 16, 9612 (2025).	Micro-LED	/	3.3 V	/	146.8 μ W
Adv. Optical Mater. 13, 2500271 (2025)	Micro-LED	/	4.3 V	/	3.6×10^4 μ W/cm ² (222 A/cm ²)
Nat. Nanotechnol. 17, 500-506 (2022).	Micro-LED	MoS ₂ transistor	8 V	8 V	~150 μ W
Light Sci. Appl. 12, 258 (2023)	Micro-LED	Si-CMOS	3.2 V	5 V	2.16 μ W (0.37 A/cm ²)
This work	Micro-LED	MoS₂ transistor	3 V	3 V	14.4 μW (95.1 A/cm²)

Corresponding changes made in the manuscript:

In response to the reviewer’s valuable comments, we have incorporated experimental approaches to reduce power consumption in the main text and fabricated a 16×16 device array, with the corresponding results now updated in **Figure S16**. Furthermore, we have supplemented **Table S2** with survey results on power consumption. We sincerely appreciate the reviewer’s insightful suggestions, which have significantly enhanced the stringency and completeness of our study.

“On the other hand, to meet the low-power requirements of edge computing applications, we further investigated the potential of high- κ dielectrics in replacing SiO₂ gate insulator for power consumption optimization (38, 39). **Figure 5a** shows the successfully fabricated 16×16 IPPMLED arrays with an increased pixel density of 336 PPI by reducing the pixel pitch. The detailed process optimization for large-scale array integration is provided in **Figure S13** and **Figure S14**, showcasing refined methods for large-area vdW material transfer and micro-LED bonding techniques. As expected, the IPPMLED device with dielectric engineering exhibits a notable reduction in the driving voltage, while maintaining good device-to-device uniformity in both V_{th} and on/off-state across 100 devices ($C_v=9.9\%$) (**Figure S15**). Correspondingly, the inherently low power consumption of the synaptic transistor translates into a great energy efficiency advantage for the integrated LED device over current display technologies (**Figure S16 and Table S2**).” on Page 10.

Reference:

1. Xu, H., et al. A low-power vertical dual-gate neurotransistor with short-term memory for high energy-efficient neuromorphic computing. *Nat. Commun.* **14**, 6385 (2023).
2. Seo, Seokho, et al. The gate injection-based field-effect synapse transistor with linear conductance update for online training. *Nat. Commun.* **13**, 6431 (2022).
3. Li, L., et al. Floating-gate photosensitive synaptic transistors with tunable functions for neuromorphic computing. *Science China Materials.* **64**, 1219-1229 (2021).
4. Xie, T., et al. Carbon nanotube optoelectronic synapse transistor arrays with ultra-low power consumption for stretchable neuromorphic vision systems. *Adv. Funct. Mater.* **33**, 2303970 (2023).
5. Liang, X., et al. Multimode transistors and neural networks based on ion-dynamic capacitance. *Nat. Electron.* **5**, 859-869 (2022).
6. Ma, M., et al. Multiplexed neurochemical transmission emulated using a dual-excitatory synaptic transistor. *npj 2D Mater. Appl.* **5**, 23 (2021).
7. Li, F., et al. An artificial visual neuron with multiplexed rate and time-to-first-spike coding. *Nat. Commun.* **15**, 3689 (2024).
8. Chen, Y., et al. All two-dimensional integration-type optoelectronic synapse mimicking visual attention mechanism for multi-target recognition. *Adv. Funct. Mater.* **33**, 2209781 (2023).
9. Han, M. J., & Tsukruk, V. V. Trainable bilingual synaptic functions in bio-enabled synaptic transistors. *ACS Nano*, **17**, 18883-18892 (2023).
10. Hao, Z., et al. Retina-inspired self-powered artificial optoelectronic synapses with selective detection in organic asymmetric heterojunctions. *Adv. Sci.* **9**, 2103494 (2022).

Reviewer #3 (Remarks to the Author):

Response:

We are grateful to the reviewer for their co-review and constructive comments. We have systematically addressed all comments and concerns from reviewers in our point-by-point responses and the revised manuscript, with key additions including the fabrication of new device arrays, the development of a display processing platform, and an expanded literature survey. We hope our effort can clearly address the concerns of Reviewer #3.

Response to Reviewer #1:

Reviewer #1 (Remarks to the Author):

The revised manuscript addresses many of the earlier concerns, with improvements in overall organization, clearer figure annotations, and added benchmarking context.

Nonetheless, several critical points remain unresolved and currently preclude a clear assessment of the hardware contribution and system-level viability.

The authors would like to express their sincere gratitude to Reviewer #1 for carefully reviewing our manuscript and providing us with valuable feedback and constructive suggestions, which have greatly enhanced the quality of this work. In response to Reviewer #1's insightful comments, additional explanations have been provided regarding **conductance programming in practical image processing applications, the deployment of input vectors and weights, and details of the neural network architecture**. Furthermore, we have supplemented the experimental data with endurance characteristics on different IPPMLED configurations (**Figure R1**) and software-hardware inference accuracy comparisons (**Figure R2**). All revisions have been carefully incorporated into the manuscript, with detailed point-by-point responses to Reviewer #1's comments provided in the revised version, where corresponding modifications are highlighted.

Comment#1

While the HfO₂ gate lowers the threshold for read/display operation, the conductance programming for in-memory computing still relies on high amplitude pulses (~30V to -35V), leaving open questions about true low-power operation. The required programming voltages also raise concerns about device lifetime. Providing endurance data under these conditions (e.g., cycles-to-failure or stability over 10³ - 10⁴ cycles) would help assess reliability.

Response:

The authors sincerely appreciate Reviewer #1's insightful comments regarding the conductance programming voltage for in-memory computing and device lifetime. We deeply regret the inadequacy in the presentation of experimental operating voltage and the insufficient elaboration of related information in the original manuscript, which may have caused confusion. Below, we provide thorough clarifications and supplementary data to address these issues.

(1) Programming voltage for in-memory computing

Building upon the insightful suggestions raised by the reviewers in the previous round of comments regarding energy-efficient device operation, we have implemented dielectric engineering optimization on the devices by replacing the SiO₂ dielectric layer with HfO₂ in the array application section to reduce the operating voltage. As demonstrated in the testing and application data of the IPPMLED array in Figure 5, the introduction of the HfO₂ dielectric layer significantly reduces the threshold voltage for

read/display operations. **Consequently, the conductance programming for in-memory computing presented in Figure 5 is performed within a programming voltage range of -4 V to 4 V .** This range is compatible with the voltage output limits ($\leq 24\text{ V}$) of our laboratory's multi-channel voltage programming system (NI-PXI 4163) and substantially mitigates the potential power consumption issues associated with high programming voltages.

Besides, it should be emphasized that SiO_2 -based devices played an irreplaceable role in our preliminary research. Specifically, in this initial proof-of-concept stage, our primary objective was to verify the feasibility of the innovative in-pixel processing architecture. Therefore, we initially adopted a 300 nm SiO_2 dielectric layer, which provides excellent color contrast for the transferred van der Waals materials and thus ensures the reliability of the fabrication process. Accordingly, the device characterization and systematic single-device tests for the newly developed IPPMLED, presented in Figure 1 through Figure 4, were conducted using the 300 nm SiO_2 -based devices, with programming operations performed in the range of around -35 V to 30 V . In contrast, the system-level operations in Figure 5 are carried out under the optimized -4 V to 4 V range.

We sincerely regret that the previous revision of our manuscript insufficiently articulated the distinction between the operating voltages and the underlying dielectric engineering in this work, which may have caused confusion. In the revised version, we have explicitly elaborated on the optimization of operating voltages in both the figure captions and a newly added section in the revised manuscript.

(2) Supplemental endurance data for HfO_2 -based devices

Regarding the device lifetime concerns raised by Reviewer #1, our original manuscript already presented the excellent long-term endurance of the SiO_2 -based IPPMLED over 10^6 cycles (Figure S6b in the manuscript). While we sincerely apologize for the omission of endurance data for HfO_2 -based IPPMLEDs, which is critical for evaluating their stability in intelligent display applications. Following the reviewer's suggestion, we have additionally conducted endurance testing on the HfO_2 -based device over 10^4 cycles (**Figure R1a**).

As presented in **Figure R1a-b**, both SiO_2 -based and HfO_2 -based IPPMLEDs demonstrate stable switching endurance within 10^4 cycles. It should be further emphasized that the off-state current magnitude of the device in the collected endurance test plots is at the $\sim\text{nA}$ level. This is attributed to the current precision limitation of the electrometer (Keysight B1530A) employed for fast pulse testing, rather than representing the intrinsic off-state current of the devices. In contrast, the transfer characteristic curves measured by the Keysight B1500A electrometer can reach a detection limit of 10^{-14} A . Therefore, to further verify the performance reliability of the devices after 10^4 cycles of testing, we additionally utilized the high-precision Keysight B1500A to compare the transfer characteristic curves before and after the endurance test. The results confirm that no significant degradation of the off-state of the device occurred (**Figure R1c-d**).

Figure R1 Endurance characteristics of the IPPMLEDs. a,b. The endurance characteristics of HfO₂-based IPPMLED (a) and SiO₂-based IPPMLED (b) over 10⁴ cycles. **c,d.** Transfer characteristic curves measured before and after the endurance tests for the (c) HfO₂-based and (d) SiO₂-based devices, showing negligible performance degradation.

Corresponding changes made in the manuscript:

In response to Reviewer #1’s comment, the revised manuscript now clearly specifies that the in-memory computing conductance programming is performed using the HfO₂-based IPPMLED, with the device configuration clearly indicated in **the captions of Figures 1–5** to avoid any potential confusion. Furthermore, the endurance data for the HfO₂-based IPPMLED over 10⁴ cycles has been incorporated into **Figure S19** to confirm its reliability under practical operating conditions.

“On the other hand, to meet the stringent power constraints of edge computing applications, we systematically explored the potential of high-κ dielectrics in replacing SiO₂ gate insulator for power consumption optimization (51,52). HfO₂, as a widely adopted high-κ dielectric material, exhibits excellent insulating properties and favorable interface characteristics, which are compatible with the designed fabrication process flow of the IPPMLEDs. **Figure 5a** shows the successfully fabricated 16×16 HfO₂-based IPPMLED arrays with an increased pixel density of 336 PPI by reducing the pixel pitch.” on Page 10.

“As illustrated in **Figure S17a**, when the gate voltage is swept from −4 V to +4 V, the device exhibits three well-defined emission regions, characterized by a non-

emissive dead zone below 0 V, an approximately linear response regime, and finally a saturation region at intermediate to higher voltages (above 3 V). Furthermore, under programming pulses within a low voltage window of ± 4 V, the device exhibits controllable conductance modulation, mirroring the versatile synaptic plasticity essential for in-memory computing (Figures S17b-e).” on Page 10.

“Moreover, the HfO₂-based IPPMLED demonstrates robust endurance, maintaining stable operation over 10⁴ cycles (Figure S19). This reliable cycling endurance is critical for subsequent practical in-situ display applications that require long-term device stability.” on Page 10.

“Furthermore, this fabrication scheme is successfully extended to fabricate IPPMLED arrays featuring a HfO₂ dielectric layer. A uniform 20 nm HfO₂ film was deposited by atomic layer deposition (ALD), a technique selected owing to its excellent conformality and precise thickness control at the atomic scale.” on Page 14.

“For hardware-accelerated neural network inference, an integrated system comprising an HfO₂-based IPPMLED array, peripheral electronics (INA226A ADC, DAC7311 DAC), and a microcontroller unit (MCU, STM32F103RCT6) is developed. The HfO₂-based IPPMLED array is configured in a matrix layout and wire-bonded onto a PCB (ASM Pacific Technology, IHAWK AERO). Programming voltages within the range of ± 4 V are supplied by an NI-PXI 4163 modular instrumentation system, integrated with a 24-channel PXI-2532B multiplexer for row and column addressing.” on Page 16.

Comment#2

Fig. 4b and Fig. S17 do not report the essential write/read pulse parameters (amplitude, width, inter-pulse interval, and related settings), leaving the methodology unclear and preventing meaningful energy or latency estimates.

Response:

We sincerely thank the reviewer for raising this important point regarding the write/read pulse parameters in Figure 4b and Figure S17, which is crucial for enhancing the experimental rigor and completeness of this study. We apologize for the omission of these essential details in the original manuscript, as they are fundamental for ensuring methodological clarity and for supporting the quantitative analysis of key metrics, such as energy consumption. As described in the energy consumption formula $E=I_{\text{peak}}\times V\times t$, where I_{peak} is the readout current, V is the readout voltage, and t is the pulse duration, the specification of pulse conditions is fundamental to any energy calculation.

Corresponding changes made in the manuscript:

To clarify the operational parameters of the devices in this work, we have supplemented the relevant figures with key pulse details. Specifically, the pulse amplitude and width are indicated in **Figure 4b** and **Figures S17b–c**, while the pulse amplitude, width, and inter-pulse interval have been detailed in **Figure S17d**. A comprehensive review was conducted across all relevant figures in the main text and Supplementary Information, with additional annotations now provided for key

experimental conditions, including amplitude, width, inter-pulse interval, and related settings.

Comment#3

Please define “post-training” and “post-development” explicitly in both the main text and the Fig. 5f caption and clarify whether the weights were obtained ex-situ (software training/simulation) and subsequently programmed into hardware. Also, since the input vector is 25×1 (from a 5×5 image) whereas the weight map is shown as 5×5 , a short explanation of how the 25×1 input corresponds to the 5×5 weight visualization would improve interpretability. (In Fig. 5f)

Response:

We sincerely appreciate the reviewer’s insightful comments on hardware deployment, which have significantly enhanced the readability and rigor of this manuscript. Corresponding supplements and revisions have been made in the text, with detailed explanations as follows:

(1) Definitions of “post-training” and “post-deployment”

We apologize for failing to provide clear definitions of these two terms in the original manuscript, which may have caused confusion for readers. As correctly noted, **the “post-training” weights refer to the ideal, quantized values obtained ex-situ from software-based neural network training.** In the main text, the term “post-development” is not used; instead, **“post-deployment” represents the actual conductance values physically programmed into the IPPMLED array using our multi-channel voltage programming system (NI-PXI 4163).** Consistent with the reviewer’s observation, the weights in this work were indeed obtained ex-situ via software training/simulation and subsequently programmed into the hardware. The caption for Figure 5f has been updated to explicitly distinguish and compare the weight distributions from these two stages, demonstrating the accuracy and reliability of our hardware mapping process.

(2) Correspondence between the input vector and the weight visualization

To address the dimensional discrepancy between the input vector and the weight representation, this work adopts a split-vector input strategy. Specifically, each 5×5 image is first flattened into a 25×1 input vector. Owing to the size limitation of the IPPMLED array, this vector is split into 5 sub-vectors, which are sequentially fed into the array for multiply-accumulate operations with the pre-deployed 5×5 weight matrix. This input-splitting strategy has been adopted in relevant research studies (e.g., Nature Communications (2025) 16:9518).

Corresponding changes made in the manuscript:

In light of the reviewer’s valuable feedback, we have updated **the caption of Figure 5f** to include the terms “post-training” and “post-deployment” for clarity. Additionally, we have provided a clear explanation regarding the process of dividing

the input vector (25×1) into sub-vectors. We believe these additions and clarifications make the relevant descriptions in the manuscript more accurate and easier to follow.

“As shown in Figure 5f, the weights obtained from ex-situ software training are extracted, quantized, and then programmed onto the IPPMLED array via non-volatile conductive states, achieving a complete correspondence with the target weight matrix.” on Page 12.

“Each input image (5×5 pixels) is flattened into a 25×1 vector, divided into five sub-vectors, and sequentially encoded into voltage signals proportional to pixel intensity for in-situ MAC operations with the mapped 5×5 weights on the array.” on Page 16.

“f. Weight deployment and hardware inference process. Input images (5×5 pixels) are flattened into 25×1 vectors, split into five sub-vectors, and sequentially processed via multiply-accumulate operations with the mapped 5×5 weights on the array. Heatmaps show the hardware conductance distribution (post-deployment) versus the quantized software weights (post-training).” on Page 31.

Comment#4

The manuscript does not clearly describe the neural network architecture (layers, activations, parameter count), the training dataset (type/size) and split, or software-hardware accuracy comparisons. Without these, reproducibility is limited and the hardware claim is difficult to interpret.

Response:

We sincerely appreciate reviewer’s comprehensive and valuable feedback regarding the neural network architecture and parameters. Although relevant details of the network architecture had been added to the Methods section in our previous response, we fully agree that a more systematic and complete presentation of the detailed parameters—as kindly advised by the reviewer—would significantly enhance the rigor and reproducibility of this work. Therefore, in accordance with the reviewer’s helpful comments, we have further supplemented the manuscript with comprehensive parameter details and comparative information.

(1) Clarifications on neural network architecture and training dataset

Specifically, our model is a **standard multi-layer perceptron (MLP) with a three-layer fully connected architecture, comprising two hidden layers (each with 128 neurons) and one output layer corresponding to 26 categories**. The total number of trainable parameters in the entire network is 23194. Each hidden layer is followed by a ReLU activation function to enhance the model’s nonlinear representation capability. The network is trained for 250 epochs using the AdamW optimizer, with cross-entropy loss minimized for classification tasks. To monitor generalization performance, the dataset is split into independent training and validation sets, and validation accuracy is recorded after each epoch. **The dataset consists of 26 categories (letters A–Z), with 1000 samples per category, totaling 26000 grayscale images**. Each character is represented as a 5×5 pixel array. To improve the model’s robustness

against image perturbations, controlled-amplitude random noise is introduced for data augmentation. All samples are normalized to a unified scale before being fed into the network.

We have reorganized and supplemented the detailed parameters of the neural network architecture and training dataset in the **Methods** section. Additionally, the number of parameters in each layer has been annotated in Figure 5e (neural network architecture diagram) for clearer illustration.

Figure R2 Confusion matrices comparing software and hardware inference accuracy. a,b. Software inference results on IPPMLED-enhanced images (a) and noisy images (b). **c,d.** Corresponding hardware inference results on IPPMLED-enhanced images (c) and noisy images (d).

(2) Supplement of software-hardware accuracy comparisons

In the original manuscript, we provided hardware inference accuracy for both noisy images and IPPMLED-enhanced images (Figures 5h and S23). However, we omitted the software inference accuracy—a critical piece of information essential for comprehensively evaluating the performance and reliability of the hardware system. Accordingly, we have now supplemented the software inference accuracy for both clean and noise-augmented images. As shown in **Figure R2**, a direct comparison reveals only a minor accuracy loss of approximately 0.1%~0.9% in hardware compared to the software benchmark, which is likely attributable to inherent device non-idealities such

as conductance fluctuation and nonlinearity. These software-hardware accuracy comparisons convincingly demonstrate the functional reliability and effectiveness of our hardware system in performing accurate neural network inference.

Corresponding changes made in the manuscript:

Following the reviewer's valuable suggestions, we have clarified the detailed parameters of the neural network architecture and training dataset in the **Methods** section to ensure experimental reproducibility and annotated the parameters in **Figure 5e**. Furthermore, we have supplemented the software inference accuracy of noisy images and IPPMLED-enhanced images in **Figure S24** to enrich the comparative analysis.

“Notably, the hardware inference accuracies exhibit only a minor degradation of 0.1%–0.9% compared to their software counterparts (**Figure S24**), which strongly validates the functional reliability of the IPPMLED-based in-situ image display system.” on Page 13.

“A multilayer perceptron (MLP) is developed for character classification using a custom dataset of 5×5 grayscale letter images. The model adopts a three-layer, fully connected architecture, which includes two hidden layers with 128 neurons each and an output layer corresponding to the 26 character categories. The total number of trainable parameters of the entire neural network amounts to 23194. A Rectified Linear Unit (ReLU) activation function follows each hidden layer to enhance the model's nonlinear representational capacity. The network is trained for 250 epochs using the AdamW optimizer, with the cross-entropy loss serving as the optimization objective. To evaluate generalization performance, the dataset is divided into separate training and validation subsets, and validation accuracy is recorded after each epoch. The dataset comprises 26000 grayscale images across 26 classes (A–Z), with 1000 samples per class. Each character is represented as a 5×5 pixel array. To improve robustness against input perturbations, controlled-amplitude random noise is applied for data augmentation. All image samples are normalized prior to being fed into the network.” on Page 15.